# RIPK1 dephosphorylation and kinase activation by PPP1R3G/PP1γ promote apoptosis and necroptosis

Jingchun Du [1,2,9], Yougui Xiang[1,3,9], Hua Liu[1,4], Shuzhen Liu [1], Ashwani Kumar[5], Chao Xing [5,6,7] & Zhigao Wang [1,8✉]

Receptor-interacting protein kinase 1 (RIPK1) is a key regulator of inflammation and cell death. Many sites on RIPK1, including serine 25, are phosphorylated to inhibit its kinase activity and cell death. How these inhibitory phosphorylation sites are dephosphorylated is poorly understood. Using a sensitized CRISPR whole-genome knockout screen, we discover that protein phosphatase 1 regulatory subunit 3G (PPP1R3G) is required for RIPK1-dependent apoptosis and type I necroptosis. Mechanistically, PPP1R3G recruits its catalytic subunit protein phosphatase 1 gamma (PP1γ) to complex I to remove inhibitory phosphorylations of RIPK1. A PPP1R3G mutant which does not bind PP1γ fails to rescue RIPK1 activation and cell death. Furthermore, chemical prevention of RIPK1 inhibitory phosphorylations or mutation of serine 25 of RIPK1 to alanine largely restores cell death in PPP1R3G-knockout cells. Finally, $Ppp1r3g^{-/-}$ mice are protected from tumor necrosis factor-induced systemic inflammatory response syndrome, confirming the important role of PPP1R3G in regulating apoptosis and necroptosis in vivo.

[1] Department of Molecular Biology, University of Texas Southwestern Medical Center, 5323 Harry Hines Boulevard, Dallas, TX 75390, USA. [2] Department of Clinical Immunology, Kingmed School of Laboratory Medicine, Guangzhou Medical University, Guangzhou, Guangdong 510182, China. [3] Caris Life Sciences, 4610 South 44th Place, Phoenix, AZ 85040, USA. [4] School of Pharmacy, Jiangxi University of Chinese Medicine, Nanchang, Jiangxi 330006, China. [5] Eugene McDermott Center for Human Growth and Development, University of Texas Southwestern Medical Center, 5323 Harry Hines Boulevard, Dallas, TX 75390, USA. [6] Department of Bioinformatics, University of Texas Southwestern Medical Center, 5323 Harry Hines Boulevard, Dallas, TX 75390, USA. [7] Department of Population and Data Sciences, University of Texas Southwestern Medical Center, 5323 Harry Hines Boulevard, Dallas, TX 75390, USA. [8] Center for Regenerative Medicine, Heart Institute, Department of Internal Medicine, University of South Florida, 560 Channelside Drive, Tampa, FL 33602, USA. [9]These authors contributed equally: Jingchun Du, Yougui Xiang. ✉email: zhigao@usf.edu

Apoptosis and necroptosis are two distinct cell death pathways with very different immune consequences[1–3]. Apoptosis results in nuclear and cytoplasmic condensation, cell shrinkage, and small apoptotic body formation, without damaging plasma membrane integrity, thus leading to little or no immune responses. In contrast, necroptosis is manifested by organelle and cell swelling and eventual cell membrane disruption, leading to the release of cellular contents including damage-associated molecular patterns (DAMPs), which induce strong immune responses. These cell death pathways are vital for development and disease, and slight perturbation might lead to severe consequences[4,5].

One of the best studied cell death inducers is the pleiotropic tumor necrosis factor (TNF), which may activate many different pathways depending on the cellular context[6,7]. Normally, TNF engages its membrane receptor TNFR1 to induce the formation of complex I, consisting of TRADD (TNFR1-associated death domain protein) and RIPK1 (receptor-interacting protein kinase 1), as well as E3 ubiquitin ligases, such as TRAF2/5 (TNF receptor-associated factor 2 and 5), cIAP1/2 (cellular inhibitor of apoptosis 1 and 2) and LUBAC (linear ubiquitin chain assembly complex)[8]. These E3 ligases catalyze many different types of ubiquitination of RIPK1 which leads to the recruitment of TAK1 (transforming growth factor-β-activating kinase 1) as well as a protein complex including NEMO (NF-κB essential modulator) and IKKα/IKKβ (inhibitor of the NF-κB kinase α/β), to activate NF-κB signaling and cell survival[9–13]. However, under some conditions, complex I is disassembled and a cytosolic cell death-inducing protein complex called complex II forms. For example, in the presence of TNF and a protein synthesis inhibitor cycloheximide (CHX), TRADD recruits FADD (Fas-associated death domain) and caspase 8 to form complex IIa, which results in caspase 8 activation and apoptosis[8]. This type of apoptosis does not need RIPK1 and is referred to as RIPK1-independent apoptosis. On the other hand, in the presence of TNF and a cIAP inhibitor Smac-mimetic (second mitochondria-derived activator of apoptosis-mimetic), or a TAK1 inhibitor 5Z-7, RIPK1 recruits FADD and caspase 8 to form complex IIb which also activates caspase 8 to induce RIPK1-dependent apoptosis (RDA)[14,15].

Complex II can be converted to yet another cell death-inducing complex called the necrosome to activate necroptosis, if caspase 8 is inhibited and RIPK3 is expressed[16–18]. The necrosome contains RIPK1, RIPK3, and MLKL (mixed lineage kinase-like protein) as well as CK1 (casein kinase 1) family proteins CK1α, δ and ε[19–21]. Phosphorylation of MLKL by RIPK3 leads to MLKL membrane translocation, oligomerization, and further polymerization, which results in membrane disruption and necroptosis[22–27]. Recently it has been proposed that there are two types of necroptosis[15]. Type I necroptosis is induced by TNF/5Z-7/Z-VAD-FMK (T/5Z-7/Z) or TNF/Smac-mimetic/Z-VAD-FMK (T/S/Z), and type II is induced by TNF/CHX/Z-VAD-FMK (T/CHX/Z). Similar to the two types of apoptosis which differ in the requirement of RIPK1, the major difference between these two types of necroptosis is RIPK1 activation. Specifically, type I necroptosis requires the activation of RIPK1 kinase activity in complex I which induces an insoluble pool of highly ubiquitinated RIPK1 (iuRIPK1) to promote necrosome formation. For type II necroptosis, RIPK1 kinase activity is only required in necrosome to phosphorylate RIPK3.

RIPK1 is one of the central players in the regulation of both cell survival and cell death[28,29]. Its ubiquitination is essential for NF-κB activation and cell survival, while its kinase activity is required for complex IIb formation to activate apoptosis, as well as necrosome formation to activate necroptosis. As a consequence, RIPK1 itself is heavily regulated at the post-translational level to facilitate its role switch between life and death. Besides many types of ubiquitination, RIPK1 is phosphorylated at multiple sites

which leads to either positive or negative regulation of its activity[30]. For example, phosphorylation of serine 166 (S166) in the kinase domain has been widely used as an autophosphorylation marker for RIPK1 activation and has been shown to be important for its activity[31–33]. Conversely, phosphorylation of many other sites inhibits RIPK1 kinase activity, serving as cell death checkpoints to ensure tight control before committing to cell death. For instance, IKK phosphorylates S25 in the kinase domain of RIPK1 to block ATP binding, leading to inhibition of RIPK1 kinase activity and subsequent cell death[34,35]. In addition, MK2 (mitogen-activated protein kinase-activated protein kinase 2), TAK1, TBK1 (TANK-binding kinase 1), and IKKε are also shown to phosphorylate RIPK1 at various sites to inhibit RIPK1 activity and cell death[36–40]. Importantly, interference to RIPK1 inhibitory phosphorylations causes dire consequences. For example, hyperactivation of RIPK1 in *Tbk1/Tak1* double heterozygous mice leads to comorbidity of amyotrophic lateral sclerosis (ALS) and frontotemporal dementia (FTD), which is attenuated by one allele of kinase-dead RIPK1[40]. Therefore, regulation of RIPK1 phosphorylation status is of vital importance for health and disease. However, little is known about how these inhibitory phosphorylation sites are dephosphorylated.

A majority of the phospho-serines and phospho-threonines in mammalian cells are dephosphorylated by protein phosphatase 1 (PP1), which regulates a broad range of cellular processes[41,42]. The PP1 holoenzyme is an obligatory heteromer composed of a PP1 catalytic subunit (PP1c) and one or two regulatory subunits, also referred to as PP1-interacting proteins (PIPs) or regulatory interactors of protein phosphatase one (RIPPOs)[43]. There are three PP1 catalytic subunits in mammalian cells, PP1α, PP1β, and PP1γ, which catalyze the dephosphorylation reaction toward a wide range of proteins in vitro, when stripped of their interacting proteins. Substrate selectivity of the PP1 holoenzyme in vivo is provided by PIPs, with each recruiting a specific subset of substrates to the catalytic subunit. There are about 200 validated PIPs, which assemble into more than 650 different PP1 holoenzymes, to achieve a broad substrate spectrum[41]. In addition to substrate recruitment, many PIPs also regulate PP1 subcellular localization. For example, PP1c is anchored to glycogen particles through seven different glycogen-targeting subunits, including protein phosphatase 1 regulatory subunit 3 A to 3G (PPP1R3A-PPP1R3G), to regulate glycogen metabolism[42].

In this report, we identified PPP1R3G as an essential gene required for necroptosis using a genome-wide CRISPR knockout screen. Furthermore, we demonstrated that PPP1R3G/PP1γ holoenzyme interacts with complex I to dephosphorylate inhibitory phosphorylation sites on RIPK1 to activate both apoptosis and necroptosis. Finally, *Ppp1r3g*^{−/−} mice are resistant to the lethal TNF-induced systematic inflammatory response syndrome (SIRS), which is a mouse model of sterile sepsis[44], confirming the essential role of PPP1R3G in cell death regulation in vivo.

## Results

**CRISPR whole-genome knockout screen in modified HAP1 cells identifies genes essential for necroptosis.** To identify novel regulators of necroptosis, we first generated a modified haploid cell line that is highly sensitive to necroptosis. Parental HAP1 cells do not express endogenous RIPK3 or MLKL (lane 2, Fig. 1a). The established HAP1:RIPK3:MLKL cell line was engineered to stably express a Tet-repressor (TetR). Additionally, it was engineered to express RIPK3 fused to a dimerization domain (DmrB) and MLKL fused to mCherry, both under a doxycycline (Dox)-inducible promoter. Dox-induced about three-fold increase of RIPK3-DmrB and MLKL-mCherry expression,

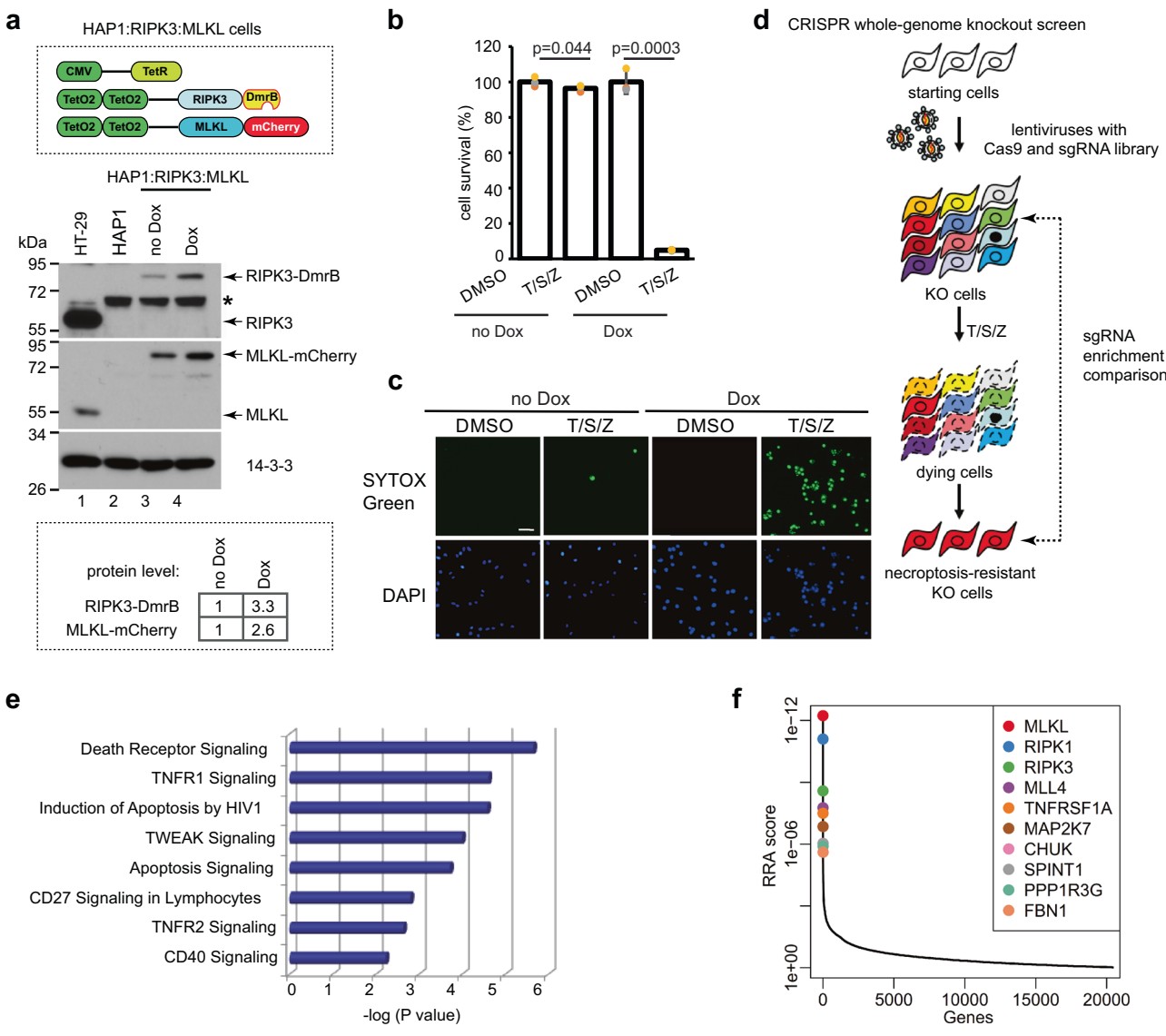

**Fig. 1 CRISPR whole-genome knockout screen in modified HAP1 cells identifies genes essential for necroptosis. a** Establishment of the HAP1:RIPK3:MLKL cell line. Upper panel, diagrams of stable transgenes in this modified HAP1 cell line. Human RIPK3 was fused to a dimerization domain called DmrB, and MLKL was fused to mCherry. Both transgenes were under the control of a tet-inducible promoter (TetO2). The cells also expressed a Tet Repressor (TetR) under the control of a CMV promoter. Middle panel, Western blotting result with antibodies against RIPK3, MLKL, and 14-3-3. Doxycycline (Dox) was used to induce transgene expression. *denotes a non-specific band. Lower panel, protein level was compared with Image J quantification. Western blotting was performed twice and representative images were shown. See Supplementary Fig. 3 for uncropped blots. **b** HAP1:RIPK3:MLKL cells were induced with or without Dox for 24 h followed by DMSO or T/S/Z treatment for 16 h. Cell survival was measured with CellTiter-Glo assay. The luminescence reading for DMSO-treated cells was assigned as 100 percent. CellTiter-Glo results are represented as mean ± SD of $n = 3$ biological independent samples. T, 20 ng/ml TNF; S, 100 nM Smac-mimetic; Z, 20 μM Z-VAD-FMK. **c** HAP1:RIPK3:MLKL cells were induced with or without Dox for 24 h followed by DMSO or T/S/Z treatment for 16 h. Cells were then stained with DAPI and a cell-impermeable DNA dye SYTOX Green. Scale bar equals 20 μm. **d** Schematic diagram of the CRISPR whole-genome knockout screen. HAP1:RIPK3:MLKL cells were transduced with lentiviruses expressing Cas9 and the sgRNA library at 0.3 multiplicity of infection (MOI), followed by 16 days of selection to obtain stable knockout cells. These cells were then treated with T/S/Z for 48 h followed by 5-day recovery. Genomic DNA from the KO cells and the necroptosis-resistant cells were used as template to amplify sgRNA sequences followed by deep sequencing. **e** Pathway analysis of the positive hits from the screen. Fisher's Exact Test was performed. $P < 0.05$. **f** Top ten positive hits from the screen according to the robust rank aggregation (RRA) score. For more screening information, see Supplementary Data and Supplementary Figs. 1 and 2.

which triggered necroptosis in almost 100% cells with treatment of T/S/Z, while little cell death was observed when Dox was omitted (Fig. 1b, c). Next, we carried out a whole-genome knockout screen in these cells using the GeCKO v2 library (Addgene, 1000000048) (Fig. 1d). Pathway analysis revealed that the necroptosis-resistant cells were highly enriched with sgRNAs against genes associated with death receptor and apoptosis signaling pathways (Fig. 1e and Supplementary Data). Known necroptosis players MLKL, RIPK1, RIPK3, and TNFR1 (gene *TNFRSF1A*) were among the top five positive hits, validating the effectiveness of the screen (Fig. 1f and Supplementary Fig. 1).

**Loss of PPP1R3G blocks T/S/Z-induced necroptosis and T/S-induced apoptosis, but not T/CHX-induced cell death.** To

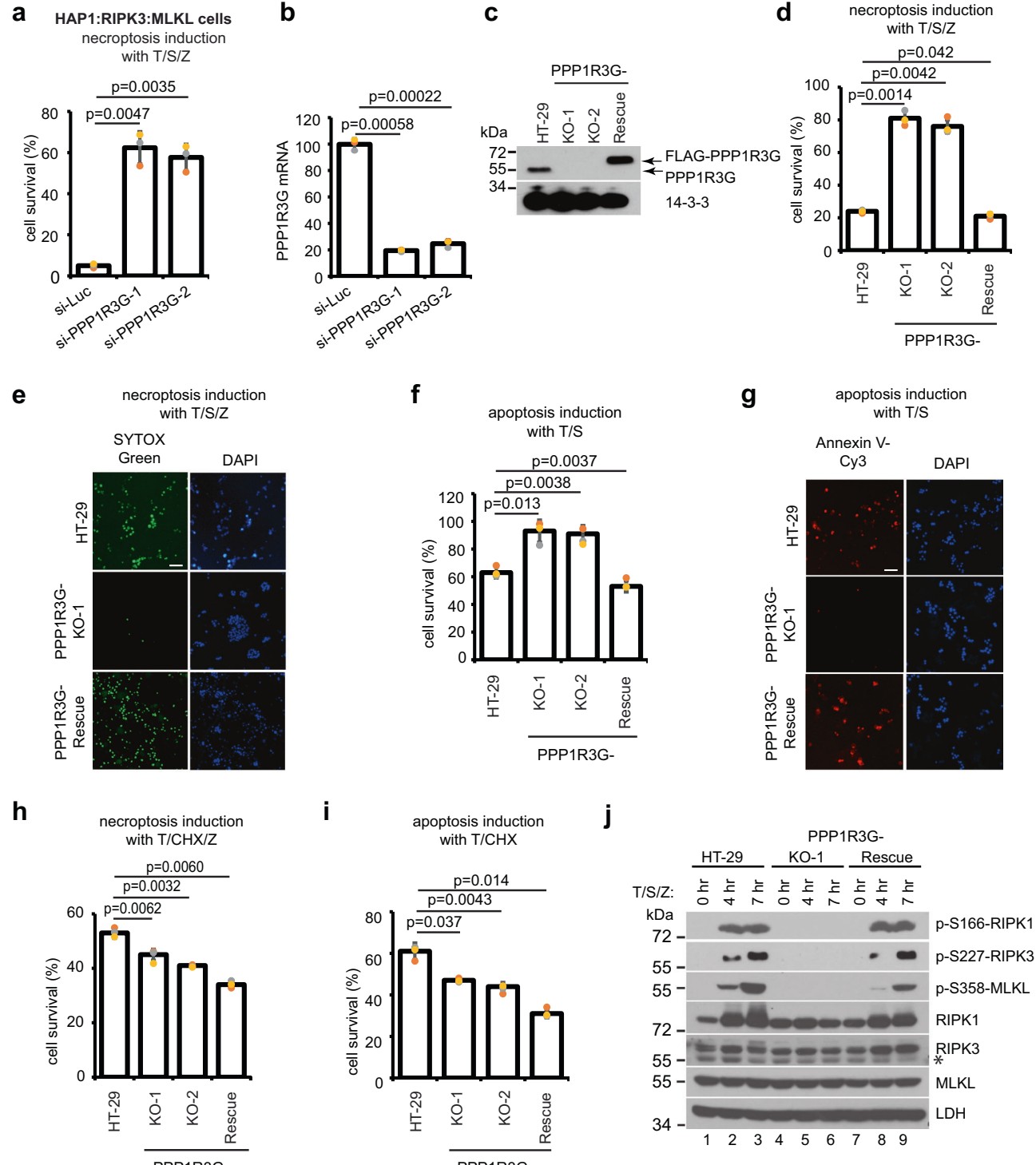

validate the hits, we performed a secondary screen using small interfering RNAs (siRNAs) targeting the top hits in HAP1:-RIPK3:MLKL cells. Among many positive hits, silencing PPP1R3G with two different oligos efficiently inhibited necroptosis (Fig. 2a, b and Supplementary Fig. 2). The same effect was observed in another cell line, human colon cancer HT-29 cells, which express endogenous RIPK3 and MLKL. Inactivation of PPP1R3G by CRISPR-Cas9 in HT-29 cells blocked T/S/Z-induced necroptosis (Fig. 2c–e). Furthermore, stably expressing a similar level of 3xFLAG-tagged PPP1R3G by lentiviral transduction in the KO cells rescued cell death effect (Fig. 2c–e). In

addition, knockout of PPP1R3G also blocked T/S-induced apoptosis, assayed by CellTiter-Glo and Annexin V staining (Fig. 2f, g). However, when a protein synthesis inhibitor CHX was used to induce necroptosis with TNF and Z-VAD-FMK, PPP1R3G-KO cells did not show any resistance. In fact, they displayed slightly more cell death than parental HT-29 cells (Fig. 2h). The same was true with T/CHX-induced apoptosis (Fig. 2i). It is known that T/CHX-induced cell death depends on complex IIa (TRADD/FADD/caspase 8) formation and does not require RIPK1, while T/S-induced RDA and T/S/Z-induced type I necroptosis depend on RIPK1 kinase activation[14,15].

**Fig. 2 Loss of PPP1R3G blocks T/S/Z-induced necroptosis and T/S-induced apoptosis. a** Small interfering RNAs (siRNA) against luciferase control or PPP1R3G were transfected into HAP1:RIPK3:MLKL cells for 48 h followed by DMSO or T/S/Z treatment for 16 h. Cell survival was measured with CellTiter-Glo assay. Viable cells expressed as a percentage of DMSO-treated cells. **b** PPP1R3G mRNA level (normalized to GAPDH) was determined by RT-qPCR. Results are represented as mean ± SD of $n = 3$ biological independent samples. **c** PPP1R3G was inactivated by CRISPR-mediated knockout in HT-29 cells to generate PPP1R3G-KO cells. Two independent KO clones were examined. Lentiviruses expressing 3xFLAG-tagged PPP1R3G were transduced into the PPP1R3G-KO-1 cells to generate the PPP1R3G-Rescue cells. Western blotting was performed with PPP1R3G and 14-3-3 antibodies. Although PPP1R3G is a 358 amino acids protein with a predicted molecular weight of 37.8 KDa, it runs around 55 kDa on an SDS-PAGE gel[45]. **d** Cells were treated with T/S/Z for 16 h and cell survival was measured with CellTiter-Glo assay. Viable cells are expressed as a percentage of DMSO-treated cells. **e** Cells were treated with T/S/Z for 16 h followed by SYTOX Green and DAPI staining. Scale bar equals 20 μm. **f** Cells were treated with T/S for 16 h and cell survival was measured with CellTiter-Glo assay. Viable cells expressed as a percentage of DMSO-treated cells. For apoptosis induction, 100 ng/ml TNF and 100 nM Smac-mimetic were used for all experiments. **g** Cells were treated with T/S for 16 h followed by Annexin V-Cy3 and DAPI staining. Annexin V binds to externalized phosphoserine on the outer plasma membrane. Scale bar equals 20 μm. **h** Cells were treated with T/CHX/Z for 16 h and cell survival was measured with CellTiter-Glo assay. Viable cells expressed as a percentage of DMSO-treated cells. CHX, 10 μg/ml cycloheximide. **i** Cells were treated with T/CHX for 16 h and cell survival was measured with CellTiter-Glo assay. Viable cells expressed as a percentage of DMSO-treated cells. **j** Cells were treated with T/S/Z for 0, 4, or 7 h. Cell lysates were subjected to Western blotting with the indicated antibodies. p-S166-RIPK1 detects phosphorylated serine 166 of RIPK1, p-S227-RIPK3 detects phosphorylated serine 227 of RIPK3 and p-S358-MLKL detects phosphorylated serine 358 of MLKL. *denotes a non-specific band. CellTiter-Glo results for **a**, **d**, **f**, **h**, **i** are represented as mean ± SD of $n = 3$ biological independent samples.

Furthermore, T/CHX/Z-induced type II necroptosis does not require RIPK1 activation in complex I[15]. These results strongly suggest that PPP1R3G functions to regulate RIPK1 kinase activation. A hallmark of RIPK1 activation is serine 166 autophosphorylation[14,31]. In PPP1R3G-KO-1 cells, T/S/Z-induced S166 phosphorylation of RIPK1 was diminished, and downstream RIPK3 and MLKL phosphorylation also disappeared (Fig. 2j), confirming that PPP1R3G is required for RIPK1 kinase activation.

**Interaction with PP1γ is required for PPP1R3G to activate RIPK1 and cell death.** To gain insight into PPP1R3G function, anti-FLAG-PPP1R3G immunoprecipitation (IP) was performed in PPP1R3G-KO-1 and PPP1R3G-Rescue cells. Not surprisingly, given that PPP1R3G has been shown to target PP1 to the glycogen particles to regulate glycogen metabolism[45], many PPP1R3G-associated proteins identified by mass spectrometry were regulators of glycogen metabolism, including glycogen phosphorylase (PYGB), hexokinase 1 (HK1), pyruvate kinase PKM and phosphoglycerate kinase 1 (PGK1). Interestingly, among the three PP1 isoforms α, β, and γ, only PP1γ was found in the PPP1R3G-associated sample (Fig. 3a). The interaction between PPP1R3G and PP1γ was confirmed by anti-FLAG IP when 3xFLAG-PPP1R3G was expressed in 293T cells (lane 5, Fig. 3b). About 70% of PIPs contain a short consensus sequence of RVXF (where X stands for any amino acid) which mediates direct interaction with PP1c[46]. This sequence is also found in PPP1R3G from aa 131 to 134 encoding RVQF. When this motif was mutated to RAQA, the mutant failed to interact with PP1γ when overexpressed in 293 T cells. In contrast, the RAQA mutant interacted with PYGB with similar affinity as the WT protein (lane 6, Fig. 3b).

Next, we wanted to determine if the phosphatase activity of the PPP1R3G/PP1γ holoenzyme is required for RIPK1-dependent cell death. Stable silencing of PP1γ by CRISPR-Cas9 led to resistance to T/S/Z-induced necroptosis (Fig. 3c). In addition, PP1γ-KD cells displayed diminished RIPK1 autophosphorylation at S166 and no MLKL phosphorylation at S358, suggesting that PP1γ is required for RIPK1 activation and cell death (Fig. 3d). However, PP1γ has a broad range of substrates and is implicated in various signaling pathways, which makes it difficult to pinpoint its specific role in RIPK1 activation. We further addressed this question by employing the RAQA mutant which could not interact with PP1γ. Stable expression of the RAQA mutant in PPP1R3G-KO-1 cells failed to rescue T/S/Z-induced necroptosis (Fig. 3e). Specifically, T/S/Z-induced S166

phosphorylation of RIPK1 and downstream MLKL phosphorylation were not rescued by the RAQA mutant (Fig. 3f), suggesting that recruitment of PP1γ is essential for RIPK1 activation and subsequent necroptosis. Similar results were obtained with T/S-induced apoptosis. RAQA mutant could not rescue T/S-induced apoptosis, assayed by cell survival (Fig. 3g), caspase 8 cleavage as well as PARP-1 cleavage (Fig. 3h), confirming the essential role of PPP1R3G/PP1γ holoenzyme in RIPK1 activation and cell death.

**PPP1R3G/PP1γ interacts with complex I and is required for complex IIb formation.** It has been reported that PP1γ associates with TRAF6 through its TRAF domain to regulate innate immune response[47]. Since TRAF domain is highly conserved among TRAF family proteins[48], we examined if PP1γ also interacted with a complex I protein TRAF2. Noticeably, 12xHis-3xmyc-PP1γ interacted with endogenous TRAF2 as well as RIPK1 only in the presence of 3xFLAG-PPP1R3G (lane 3 and 4, Fig. 4a), suggesting that PPP1R3G is the mediator that recruits PP1γ to TRAF2 and RIPK1, which are components of complex I. It is also worth noticing that co-expression of PPP1R3G raised PP1γ protein level, suggesting a stabilization effect. Importantly, in PPP1R3G-Rescue cells, PPP1R3G/PP1γ holoenzyme interacted with TRAF2 and RIPK1 under basal condition (lane 3, Fig. 4b), and the association of PPP1R3G/PP1γ with RIPK1 increased dramatically upon T/S/Z treatment (lane 4, Fig. 4b). To further investigate the interaction landscape of endogenous PPP1R3G/PP1γ, FLAG-TNF was precipitated to isolate complex I under different conditions. In HT-29 cells, FLAG-TNF treatment induced complex I formation (lane 10, Fig. 4c), and co-treatment of S/Z with FLAG-TNF resulted in PPP1R3G/PP1γ association with complex I (lane 11, Fig. 4c). Specially, phosphorylation of RIPK1 at S166 was only detected in complex I with T/S/Z treatment. In contrast, T/CHX/Z treatment did not induce S166 phosphorylation of RIPK1 (lane 4 and lane 12, Fig. 4c), confirming the previous finding that RIPK1 is not activated in complex I for T/CHX/Z-induced type II necroptosis[15]. Importantly, in PPP1R3G-KO-1 cells, PP1γ no longer bound complex I upon T/S/Z treatment and RIPK1 was not activated (lane 15, Fig. 4c). These results suggested that Smac-mimetic induces PPP1R3G/PP1γ association with complex I to activate RIPK1 kinase activity.

Next, we examined complex IIb formation. In HT-29 cells, T/S/Z induced complex IIb assembly which led to caspase 8 oligomerization and self-cleavage to produce p43/p41 fragments (arrows) (lane 2, Fig. 4d). However, in PPP1R3G-KO-1 cells, T/S/Z treatment did not result in caspase 8 self-cleavage (lane 4,

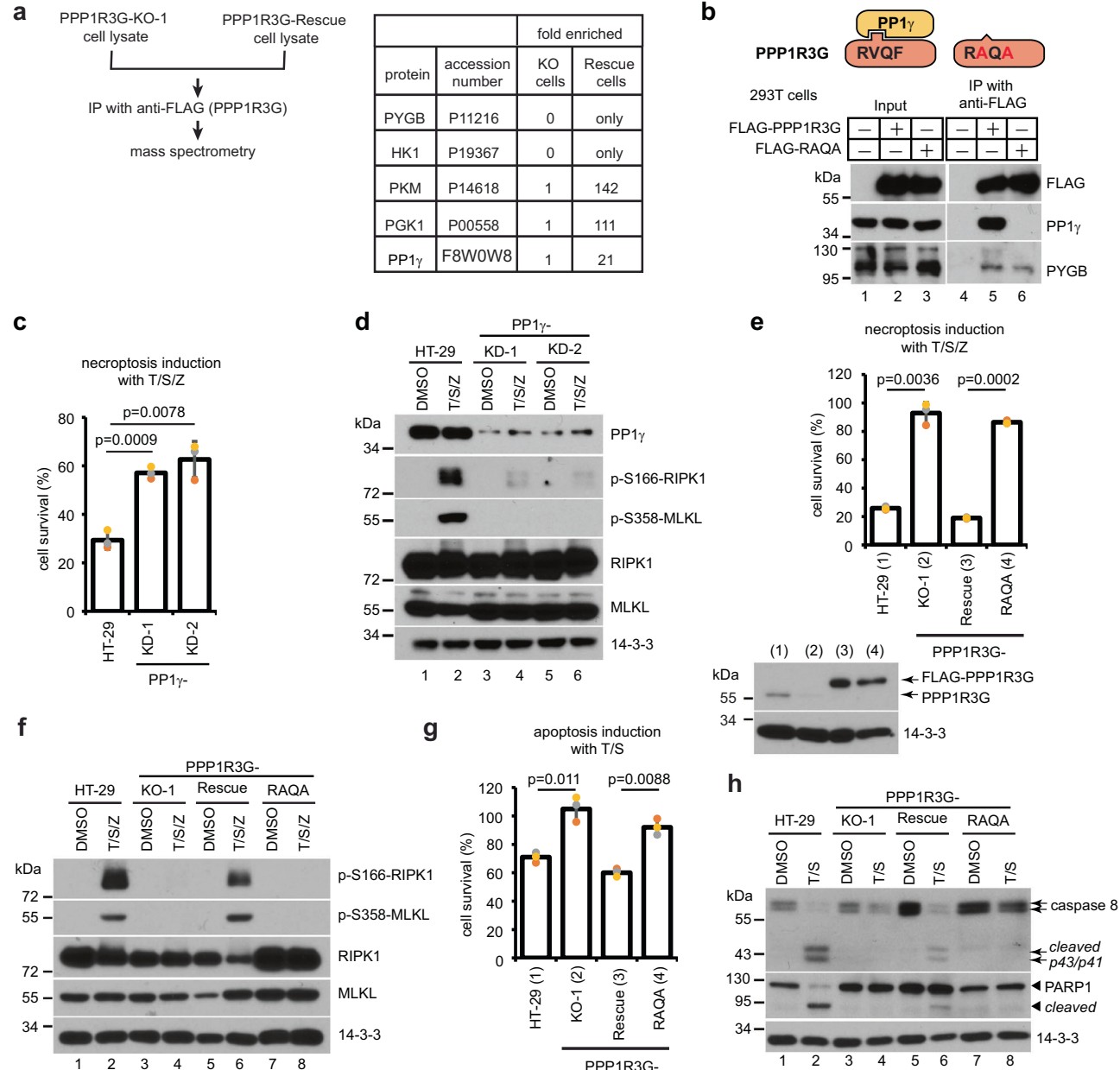

**Fig. 3 Interaction with PP1γ is required for PPP1R3G to activate RIPK1 and cell death. a** Lysates from PPP1R3G-KO-1 and PPP1R3G-Rescue cells were subjected to anti-FLAG immunoprecipitation (IP) followed by mass spectrometry analysis. Proteins highly enriched in PPP1R3G-Recue cells were listed. PYGB, glycogen phosphorylase; HK1, hexokinase-1; PKM, pyruvate kinase; PGK1, phosphoglycerate kinase 1. **b** Upper panel, diagram of the RAQA mutant. Lower panel, anti-FLAG IP result from 293T cells expressing 3xFLAG-PPP1R3G or 3xFLAG-PPP1R3G-RAQA. Western blotting was performed with FLAG, PP1γ and PYGB antibodies. **c** PP1γ was inactivated by CRISPR-Cas9 method in HT-29 cells. These cells were PP1γ-knockdown (KD) cells because there was residual PP1γ expression. Cells were treated with T/S/Z for 16 h and cell survival was measured with CellTiter-Glo assay. Viable cells expressed as a percentage of DMSO-treated cells. **d** Cells were treated with DMSO or T/S/Z for 4 h and Western blotting was performed with the indicated antibodies. **e** Lentiviruses expressing 3xFLAG-tagged PPP1R3G-RAQA were transduced into the PPP1R3G-KO-1 cells to establish the PPP1R3G-RAQA cell line. Cells were treated with T/S/Z for 16 h and cell survival was measured with CellTiter-Glo assay. Viable cells expressed as a percentage of DMSO-treated cells. Lower panel, Western blotting was performed with PPP1R3G and 14-3-3 antibodies. **f** Cells were treated with DMSO or T/S/Z for 4 h and Western blotting was performed with the indicated antibodies. **g** Cells were treated with T/S for 16 h and cell survival was measured with CellTiter-Glo assay. Viable cells expressed as a percentage of DMSO-treated cells. **h** Cells were treated with DMSO or T/S for 6 h and Western blotting was performed with indicated antibodies. Arrows denote full-length and cleaved caspase 8. Arrowheads denote full-length and cleaved PARP1. CellTiter-Glo results for **c**, **e**, and **g** are represented as mean ± SD of n = 3 biological independent samples.

Fig. 4d), suggesting that PPP1R3G acts upstream of caspase 8 activation. Indeed, IP with a caspase 8 antibody revealed that caspase 8 did not form complex IIb with FADD or RIPK1 in PPP1R3G-KO-1 cells (lane 8, Fig. 4d), confirming the essential role of PPP1R3G in complex IIb formation. Taken together, these

results cooperated for the following working model: TNF treatment induces complex I formation, and co-treatment of Smac-mimetic leads to the recruitment of PPP1R3G/PP1γ to complex I, which somehow results in RIPK1 kinase activation and subsequent complex IIb formation (Fig. 4e).

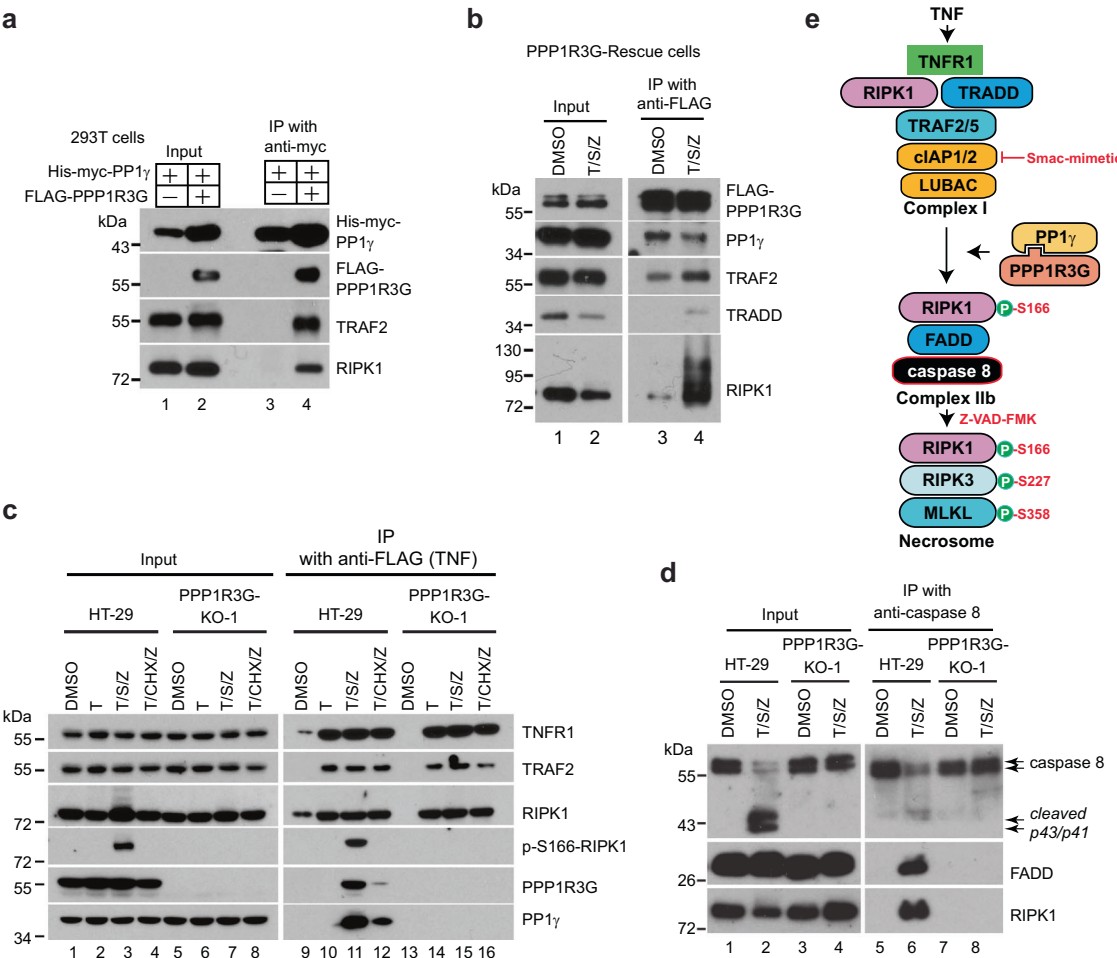

**Fig. 4 PPP1R3G/PP1γ interacts with complex I and is required for complex IIb formation. a** 3xFLAG-tagged PPP1R3G and 12xHis-3xmyc-tagged PP1γ were overexpressed in 293T cells, and cell lysates were subjected to anti-myc IP. Western blotting was performed with antibodies against myc, FLAG, TRAF2, and RIPK1. **b** PPP1R3G-Rescue cells were treated with DMSO or T/S/Z for 4 h. Cell lysates were subjected to anti-FLAG IP and Western blotting was performed with indicated antibodies. **c** HT-29 and PPP1R3G-KO-1 cells were treated with FLAG-TNF, FLAG-TNF/S/Z, or FLAG-TNF/CHX/Z for 1 h. Cell lysates were subjected to precipitation with anti-FLAG beads. Input and beads-associated proteins were subjected to Western botting with the indicated antibodies. **d** HT-29 and PPP1R3G-KO-1 cells were treated with T/S/Z for 4 h. Cell lysates were subjected to anti-caspase 8 IP. Western blotting was performed with indicated antibodies. Arrows denote full-length and cleaved caspase 8. Z-VAD-FMK inhibits activated caspase 8 but does not block caspase 8 self-cleavage. **e** Working hypothesis. TNF induces complex I formation. Co-treatment of Smac-mimetic leads to recruitment of PPP1R3G and PP1γ to complex I, which is required for RIPK1 kinase activation and subsequent complex IIb formation. IP experiments in **a**–**d** were repeated at least twice and representative images were shown.

**PPP1R3G/PP1γ directly removes inhibitory phosphorylations of RIPK1 to activate RIPK1 and cell death**. Many serine and threonine sites on RIPK1 are phosphorylated to inhibit RIPK1 activity. For example, IKK phosphorylates serine 25 in the ATP binding motif; TAK1 and p38/MK2 phosphorylate serine 320 and serine 335 in the intermediate domain; and TBK1/IKKε phosphorylates threonine 189 in the kinase domain[33–40] (Fig. 5a). We first examined the activity of these kinases. TNF-induced phosphorylation and degradation of IκBα, as well as p38 phosphorylation, were not affected in PPP1R3G-KO-1 cells, suggesting that PPP1R3G does not regulate IKK and MK2/p38 activation (Fig. 5a). This prompted us to hypothesize that PPP1R3G/PP1γ directly dephosphorylates the inhibitory phosphorylation sites on RIPK1 to allow RIPK1 to be activated. This hypothesis predicts that if inhibitory phosphorylations are prevented, PPP1R3G/PP1γ activity will not be required to activate cell death. Indeed, when p38 or IKK was inhibited with specific compounds, a significant amount of necroptosis was induced in PPP1R3G-KO-1 cells with T/S/Z-treatment, although the level of cell death did not match T/S/Z-treated HT-29 cells, suggesting that multiple kinases inhibit cell death simultaneously. As expected, these inhibitors also enhanced cell death in HT-29 cells (Fig. 5b). At the molecular level, inhibiting IKK induced S166 autophosphorylation of RIPK1 and downstream MLKL phosphorylation in T/S/Z treated PPP1R3G-KO-1 cells. However, the level was significantly lower than in T/S/Z-treated HT-29 cells, indicating that other kinases besides IKK also contribute to inhibitory phosphorylations (Fig. 5c). Taken together, these results suggest that if inhibitory phosphorylations are prevented, RIPK1 can be activated in PPP1R3G-KO-1 cells to promote a significant amount of cell death.

Next, we wanted to determine if PP1γ could directly dephosphorylate RIPK1. S320 phosphorylation of RIPK1 reached a high level at 15 min after T/S/Z treatment and decreased drastically at 4 h, at which time S166 phosphorylation of RIPK1 reached high level (Fig. 5d), confirming previously reported reverse correlation between S320 and S166 phosphorylation[36]. In an in vitro assay, phospho-S320 was readily dephosphorylated by recombinant

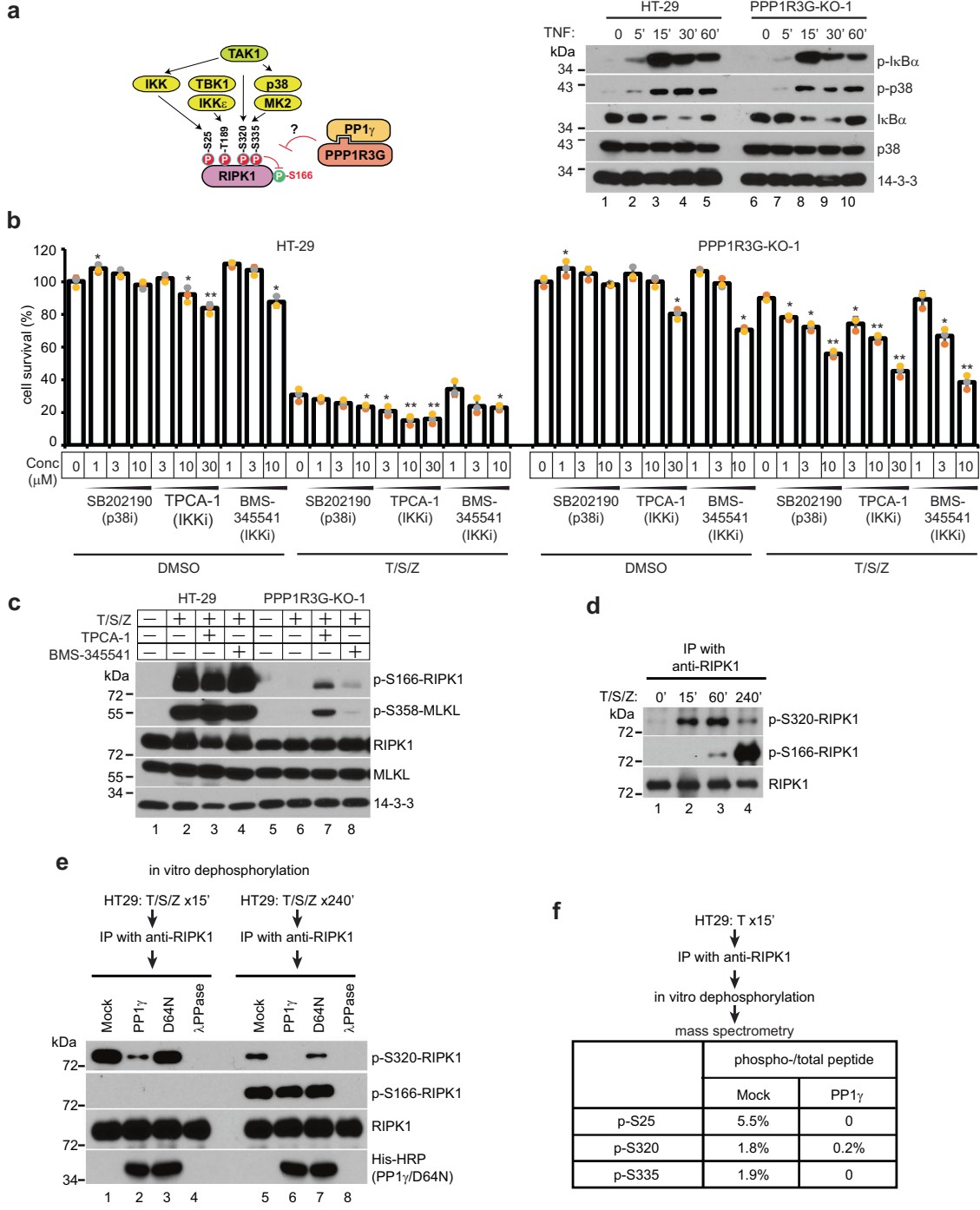

**Fig. 5 PPP1R3G/PP1γ directly removes inhibitory phosphorylations of RIPK1 to activate RIPK1 and cell death. a** Left panel, diagram of how RIPK1 kinase activity is regulated by inhibitory phosphorylations. Serine 166 is an auto-phosphorylation site of RIPK1, serving as a readout of RIPK1 kinase activation. Right panel, cells were treated with TNF for different durations and cell lysates were subjected to Western blotting with indicated antibodies. p-IκBα detects phosphorylated serine 32 of IκBα, and p-p38 detects phosphorylated Thr180/Tyr182 of p38. **b** HT-29 and PPP1R3G-KO-1 cells were treated with DMSO or T/S/Z, in the presence of an increasing concentration of different inhibitors, followed by CellTiter-Glo assay. Viable cells expressed as a percentage of DMSO-treated cells. SB202190 is a p38 inhibitor, and TPCA-1, BMS345541 are IKK inhibitors. Results are represented as mean ± SD of $n = 3$ biological independent samples. * $t$-test, $P < 0.05$; ** $t$-test, $P < 0.01$. **c** HT-29 and PPP1R3G-KO-1 cells were treated with DMSO, T/S/Z, or T/S/Z with 30 μM TPCA-1 or 10 μM BMS345541 for 4 h. Western blotting was performed with the indicated antibodies. **d** HT-29 cells were treated with T/S/Z for indicated periods and cell lysates were harvested and subjected to IP with an anti-RIPK1 antibody, followed by Western blotting. p-S320-RIPK1 detects phosphorylated S320 of RIPK1. **e** HT-29 cells were treated with T/S/Z for 15 min or 240 min and cell lysates were subjected to anti-RIPK1 IP. RIPK1-bound beads were then incubated with recombinant PP1γ, D64N mutant of PP1γ, or lambda phosphatase (λPPase), followed by Western blotting. **f** In vitro dephosphorylation was performed as in (**e**) and the final products were subjected to mass spectrometry analysis. The percentage of each phospho-peptide was shown.

PP1γ, but not an enzyme-dead mutant D64N. Interestingly, phospho-S166 was not efficiently dephosphorylated by PP1γ, while λPPase wiped out S166 phosphorylation (Fig. 5e). These results suggest that PP1γ might selectively dephosphorylate inhibitory phosphorylation sites on RIPK1. Indeed, mass spectrometry analysis revealed that recombinant PP1γ directly dephosphorylated S25, S320, and S335 phosphorylation of RIPK1 (Fig. 5f).

**RIPK1 mutants with mutated inhibitory phosphorylation sites activate cell death without PPP1R3G.** To determine which inhibitory sites were important for PPP1R3G/PP1γ-dependent activation of RIPK1, serine to alanine mutant of RIPK1 was stably expressed in the PPP1R3G/RIPK1 double knockout (DKO) cells. Similar to PPP1R3G-KO cells, DKO:RIPK1-WT cells were resistant to T/S/Z-induced necroptosis. In comparison, DKO:RIPK1-S25A cells largely restored cell death (Fig. 6a). RIPK1-S320A/S335A mutant but not S320A alone also restored significant amount of cell death, although the levels were appreciably less than S25A mutant. Surprisingly, the S25A/S320A/S335A mutant did not shown any cell death induction. It is likely that the triple mutations impair RIPK1 function. Indeed, when WT and mutant RIPK1 were expressed in 293T cells, S25A/S320A/S335A only displayed 10% of the auto-phosphorylation activity comparing to WT, while S25A and S320A/S335A retained 60 and 50% activity, respectively (Fig. 6b). These results suggest that S25 is one of the important phosphorylation sites for inhibiting RIPK1 activation, and phosphorylation of S320 and S335 also contributes to the inhibition. T189A mutant was not included, because it behaves as a dead kinase[40].

Correlating with the cell death result, phospho-S166 of RIPK1 was much stronger in DKO:RIPK1-S25A cells than in DKO:RIPK1-WT cells (Fig. 6c). Interestingly, phospho-MLKL signal was stronger at 4 h in DKO:RIPK1-S25A cells than in HT-29 cells, but it did not reach the same intensity at 7 h. This suggests that RIPK1-S25A could be activated at an earlier time point than the WT, but could not reach the full extent of activation, again indicating that there are other phosphorylations sites besides S25 that negatively regulate RIPK1. Similar results were obtained with T/S-induced RDA. DKO:RIPK1-S25A cells showed more apoptosis than DKO:RIPK1-WT cells, but less cell death than HT-29 cells, assayed with cell survival and caspase 8 cleavage, as well as PARP-1 cleavage (Fig. 6d, e). These data prompted us to propose the following working model. TNF induces complex I formation and RIPK1 ubiquitination, leading to activation of a series of kinases, which phosphorylate RIPK1 at multiple sites to inhibit RIPK1 kinase activity and cell death. In the presence of Smac-mimetic, PPP1R3G/PP1γ is recruited to complex I and dephosphorylates the inhibitory phosphorylation sites on RIPK1 to allow RIPK1 activation, leading to complex IIb formation and subsequent apoptosis. When caspase 8 is inhibited by Z-VAD-FMK, type I necroptosis is induced (Fig. 6f).

***Ppp1r3g*** ⁻/⁻ **mice are protected from TNF-induced systematic inflammatory response syndrome (SIRS).** It has been shown that *Ppp1r3g*-deficient mice are viable and have less glycogen deposition in liver. They also have reduced body weight and fat composition upon high-fat diet feeding[49]. But the in vivo role of Ppp1r3g in cell death regulation has not been investigated. To this end, we generated knockout mice using CRISPR-Cas9 method to delete most of the coding region (Fig. 7a–c). Similar to human cells, mouse embryonic fibroblasts (MEF) derived from *Ppp1r3g* ⁻/⁻ mice were more resistant to T/S/Z-induced type I necroptosis and T/S-induced RDA than those from WT litter mates, but no obvious difference was observed for T/CHX-induced apoptosis or T/CHX/

Z-induced type II necroptosis (Fig. 7d). Specifically, in WT MEF cells, T/S/Z-induced S321 phosphorylation of RIPK1 (corresponding to S320 of human RIPK1) reached a high level at 120 min and decreased afterwards, leading to activation of MLKL phosphorylation. However, in *Ppp1r3g* ⁻/⁻ MEF cells, T/S/Z-induced S321 phosphorylation of RIPK1 continued to accumulate, resulting in inhibition of MLKL phosphorylation (Fig. 7e). These results confirm that without PPP1R3G, inhibitory phosphorylation at S321 is not dephosphorylated to activate RIPK1 to induce downstream cell death.

SIRS is an excessive defense response of the body to insults, including infection, trauma, acute inflammation, and ischemia. TNF-induced SIRS is a mouse model of sterile sepsis[44]. It is widely used in recent times to evaluate the in vivo function of necroptosis pathway components, including RIPK3, RIPK1, caspase 8, and MLK[50–52]. Injection of mouse TNF resulted in 100% mortality in 24 h in WT mice. However, about 80% of *Ppp1r3g* ⁻/⁻ mice survived (Fig. 7f). The body temperature of WT mice decreased all the way to about 25 °C after TNF injection and never recovered before death, while the temperature of KO mice dropped to about 32 °C at 8 h and then started to recover until becoming normal at 24 h (Fig. 7g). Moreover, the serum cytokine levels, including IL-6, CXCL1, and CXCL2, were significantly lower in KO mice than in WT mice (Fig. 7h), indicating reduced inflammation. Hematoxylin/eosin (H&E) staining revealed that the ileum of WT mice after TNF injection was severely damaged (panel III of Fig. 7i) comparing to the relatively normal appearance of KO mice (panel IV). Injection of TNF also induced hepatocyte necrosis in WT mice, leading to regions of vacant space from hepatocyte dropout (*, panel VII of Fig. 7i). Around those regions, clusters of red bloods cells could be found, suggesting that blood vessels were permeabilized due to inflammation (arrows, panel VII), while the KO mice generally did not display these changes (panel VIII). At the molecular level, phosphorylation of RIPK1-S321 was elevated and phosphorylation of MLKL-S345 was diminished in KO liver after TNF injection (Fig. 7j) These results indicate that loss of Ppp1r3g protects mice from systemic inflammation caused by TNF.

## Discussion

RIPK1 is a key regulator of cell survival, inflammation, and cell death. Importantly, RIPK1 kinase activity is required for apoptosis and necroptosis, depending on cellular context. Therefore, RIPK1 is heavily regulated post-translationally to ensure tight control over its activity. In particular, many sites on RIPK1 are phosphorylated to inhibit its kinase activity, serving as functional cell death checkpoints. How these inhibitory phosphorylations are reversed to allow RIPK1 kinase activation is poorly understood. In this report, we identified a phosphatase subunit PPP1R3G that was essential for T/S/Z-induced necroptosis from a sensitized whole-genome CRISPR knockout screen. Loss of PPP1R3G blocked T/S-induced RDA and T/S/Z-induced type I necroptosis, both of which depend on RIPK1 kinase activity (Fig. 2a–f). However, PPP1R3G was not required for T/CHX-induced RIPK1-independent apoptosis or T/CHX/Z-induced type II necroptosis which does not require RIPK1 activation in complex I (Fig. 2h, i). These results strongly suggest that PPP1R3G promotes RIPK1 activation-dependent cell death. This was further supported by the observations that PPP1R3G and its associated catalytic subunit PP1γ were recruited to complex I and is required for RIPK1 activation and complex IIb formation to induce cell death (Figs. 3 and 4).

Multiple pieces of evidence suggest that PPP1R3G/PP1γ holoenzyme dephosphorylates inhibitory phosphorylations sites on RIPK1 to allow RIPK1 kinase activation. First, in PPP1R3G-

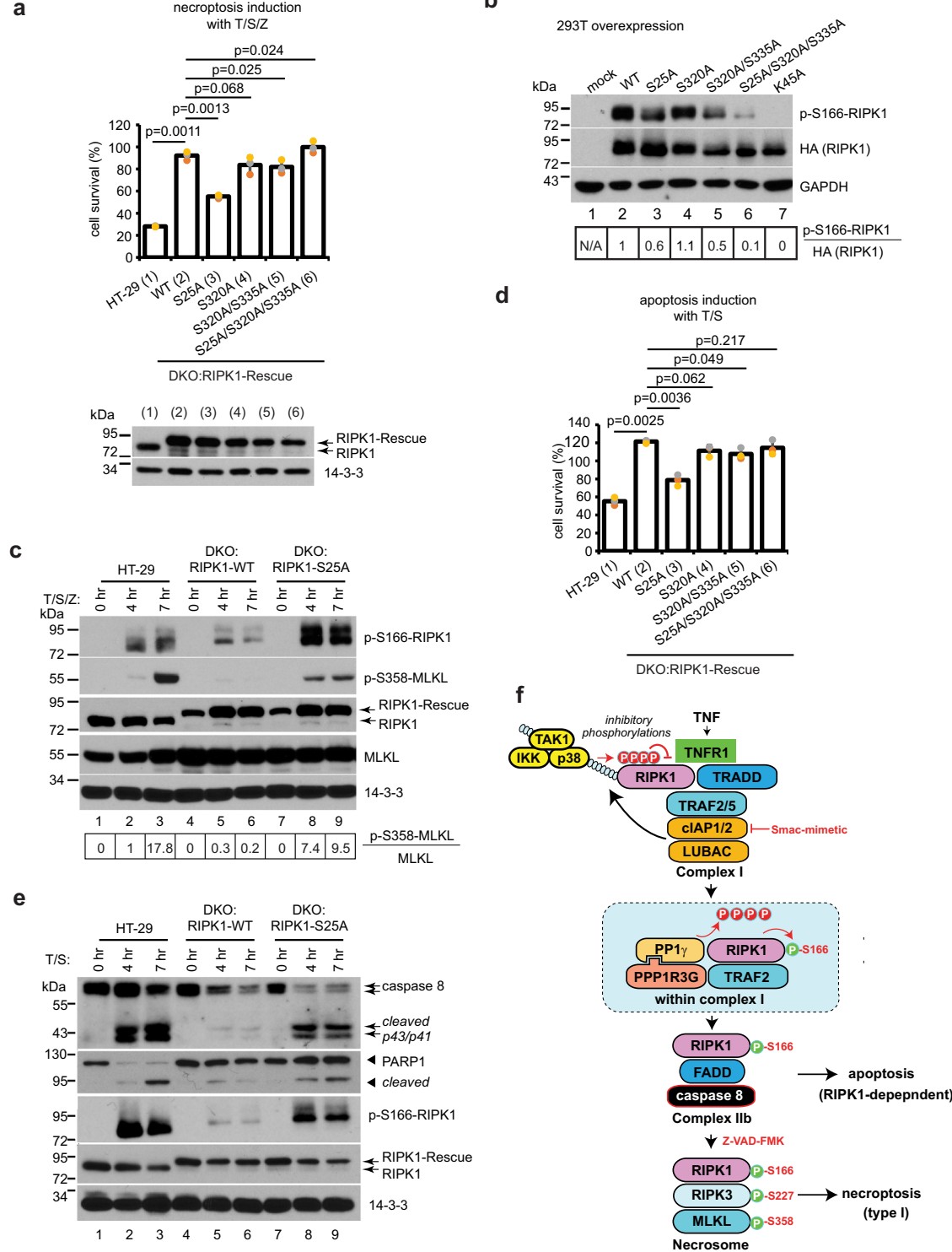

KO and PP1γ-KD cells, RIPK1 was not activated to autophosphorylate S166. Second, a PPP1R3G-RAQA mutant that could not bind PP1γ was not able to restore RIPK1-S166 phosphorylation and cell death. Furthermore, chemical prevention of inhibitory phosphorylations partially restored RIPK1 kinase activity and cell death in PPP1R3G-KO cells, suggesting that PPP1R3G/PP1γ holoenzyme counters inhibitory phosphorylations to activate RIPK1. Importantly, recombinant PP1γ directly dephosphorylated S25, S320, and S335 phosphorylation of RIPK1. Finally, RIPK1-S25A mutant largely restored RIPK1 kinase activation and cell

death in PPP1R3G-deficient cells, and RIPK1-S320A/S335A mutant also restored a significant amount of cell death (Fig. 6), suggesting that multiple phosphorylation sites function together to inhibit RIPK1 activity. It has been suggested that MK2 phosphorylates S320 of RIPK1 in both complex I and cytosol[36]. Our results show that PPP1R3G/PP1γ associates with RIPK1 in complex I upon T/S/Z treatment (Fig. 4c). In addition, there is appreciable TRAF2/RIPK1 interaction with PPP1R3G/PP1γ at basal level (lane 3, Fig. 4b), suggesting that PPP1R3G/PP1γ might also dephosphorylate cytosolic pool of RIPK1. All these complex

**Fig. 6 RIPK1 mutants with mutated inhibitory phosphorylation sites activate cell death without PPP1R3G. a** 3xHA-tagged RIPK1-WT, RIPK1-S25A, RIPK1-S320A, RIPK1-S320A/S335A, or RIPK1-S25A/S320A/S335A was stably expressed by lentiviral transduction in the PPP1R3G/RIPK1 double knockout (DKO) cells to establish the DKO:RIPK1-Rescue cell lines. Top panel, cells were treated with T/S/Z for 16 h followed by CellTiter-Glo assay. Viable cells expressed as a percentage of DMSO-treated cells. Lower panel, cell lysates were subjected to Western blotting with RIPK1 and 14-3-3 antibodies. **b** 3xHA-tagged RIPK1-WT or mutants were expressed in 293T cells and cell lysates were subjected to Western blotting with indicated antibodies. Band intensity was measured with Image J and the ratio of p-S166-RIPK1 over total HA-RIPK1 was shown at the bottom, with RIPK1-WT assigned as 1. **c** Cells were treated with T/S/Z for 0, 4, or 7 h and cells lysates were subjected to Western blotting with the indicated antibodies. Band intensity was measured with Image J and the ratio of p-S358-MLKL over total MLKL was shown at the bottom, with HT-29 cells treated with T/S/Z for 4 h assigned as 1. **d** Cells were treated with T/S for 16 h followed by CellTiter-Glo assay. Viable cells expressed as a percentage of DMSO-treated cells. **e** Cells were treated with T/S for 0, 4 or 7 h and cells lysates were subjected to Western blotting with the indicated antibodies. **f** Working model. TNF induces complex I formation and RIPK1 poly-ubiquitination. Ubiquitinated RIPK1 recruits numerous kinases, which phosphorylate RIPK1 on multiple sites to inhibit RIPK1 kinase activity and cell death. Smac-mimetic treatment results in recruitment of PPP1R3G/PP1γ to complex I. PP1γ then dephosphorylates the inhibitory phosphorylation sites on RIPK1 to activate RIPK1 kinase activity, leading to complex IIb formation and cell death. CellTiter-Glo results for a and d are represented as mean ± SD of $n = 3$ biological independent samples.

mechanisms are intertwined in the regulation of RIPK1 kinase activity, highlighting its importance for cell death and inflammation. It is also conceivable that PPP1R3G/PP1γ might dephosphorylate other important cell death regulators besides RIPK1 to activate cell death. One piece of evidence is that DKO:RIPK1-S25A cells displayed higher phospho-S166 but did not activate phospho-MLKL or cell death to the same extent comparing to parental cells (Fig. 6c). Future phospho-proteomics studies might help answer this interesting question. It is worth mentioning that another phosphatase PPM1B (protein phosphatase 1B, or protein phosphatase 2C isoform beta) also regulates necroptosis. Opposite to RIPK1 dephosphorylation by PPP1R3G/PP1γ, dephosphorylation of RIPK3 by PPM1B inhibits RIPK3 activity and cell death[53].

PPP1R3G was previously shown to be a glycogen targeting subunit for PP1α and PP1β[45]. It is highly induced in the liver during fasting-refeeding cycle to activate glycogen synthesis[45,54]. It is interesting that only PP1γ was identified in our FLAG-PPP1R3G IP experiment. It is possible that PPP1R3G targets PP1α and PP1β to glycogen particles, while it targets PP1γ to complex I, so that one regulatory subunit could target different catalytic subunits to modulate different cellular processes. Furthermore, PP1γ did not interact with TRAF2 or complex I without PPP1R3G (Fig. 4a, c). Therefore, PPP1R3G is the subunit that brings PP1γ to TRAF2 and then complex I. The TRAF domain that mediates the interaction between PP1γ complex and TRAF2 or TRAF6 is conserved among TRAF1 to TRAF6[48]. It is conceivable that PPP1R3G/PP1γ holoenzyme could be directed to different subsets of substrates by different TRAF proteins, thus greatly increasing the substrate repertoire. These TRAF proteins might also perform redundant function so that important cellular processes such as cell death can sustain impairment to the function of one protein. In addition, there is redundancy among PP1 catalytic subunits as well[55]. For example, both PP1α and PP1γ, but not PP1β, dephosphorylate RNA sensors RIG-I (Retinoic Acid-Inducible Gene I) and MDA-5 (Melanoma Differentiation-Associated Protein 5) to activate antiviral innate immunity[56]. Our discovery adds another exciting chapter to the ever-fascinating PP1 story.

It was reported that $Caspase\ 8^{+/-}/Mlkl^{-/-}$ mice were better protected from TNF-induced SIRS than $Caspase\ 8^{+/-}$ mice or $Mlkl^{-/-}$ mice, suggesting that both caspase 8-dependnet apoptosis and MLKL-dependent necroptosis are important for the pathogenesis of TNF-induced SIRS[51]. $Ppp1r3g^{-/-}$ mice were strongly protected from TNF-induced SIRS, confirming the essential function of PPP1R3G in regulating apoptosis and necroptosis in vivo and suggesting that PPP1R3G/PP1γ represents a potential therapeutic target for systemic inflammatory disorders. RIPK1-dependent cell death also contributes to

ischemia-reperfusion-induced cardiac and kidney injury, acute pancreatitis as well as viral infection[5]. It will be interesting to examine how $Ppp1r3g^{-/-}$ mice behave in those mouse models.

Human genetics reveals that perturbation of RIPK1 signaling leads to many severe inflammatory and degenerative diseases, including autoinflammatory disorders and ALS. For instance, patients with biallelic $RIPK1$ loss-of-function mutations suffer recurrent infections, early-onset inflammatory bowel disease, and progressive polyarthritis[57,58]. Moreover, $RIPK1$ point mutations that are resistant to caspase cleavage lead to autosomal dominant autoinflammatory diseases[59,60]. Furthermore, human $TBK1$ mutations that cause elevated RIPK1 activity and neuroinflammation in the central nerve system result in ALS/frontotemporal dementia (FTD) comorbidity. Importantly, $Tbk1^{+/-}/Tak1^{+/-}$ double heterozygous mice develop ALS/FTD symptom which is alleviated by RIPK1 deficiency[40]. Therefore, RIPK1 and its regulators are important drug targets for intervening these devastating diseases[29]. Our report suggests that PPP1R3G/PP1γ will be another important therapeutic target for these RIPK1 perturbation-associated human diseases. Disruption of PPP1R3G/PP1γ interaction or PPP1R3G/TRAF2 interaction will potentially ameliorate diseases caused by elevated RIPK1 activity.

## Methods

Mouse work described in this manuscript has been approved and conducted under the oversight of the Institutional Animal Care and Use Committee at UT Southwestern. Recombinant DNA and mammalian cell line studies comply with all relevant ethical regulations supervised by the Institutional Biosafety Committee at UT Southwestern.

**General reagents.** The following reagents and antibodies were used: ZVAD-FMK, cycloheximide, SB202190, TPCA-1, BMS-345541 (ApexBio), SYTOX Green and DAPI (Invitrogen), Annexin V-Cy3 staining kit (BioVision), mouse IL-6 ELISA kit (Enzo, ADI-900-045), mouse CXCL1 ELISA kit (R&D, DY453-05), mouse CXCL2 ELISA kit (R&D, DY452-05), anti-FLAG M2 (F1804, 1 μg/ml for IP,) and anti-myc (A7470, 1 μg/ml for IP) antibody and affinity gel (Sigma), anti-RIPK1 (BD551042, 1:1000; 1 μg/ml for IP), anti-phospho-S166 of RIPK1 (Cell Signaling, 65746S, 1:1000), anti-phsopho-S320 of RIPK1 (gift from Dr. Pascal Meier, 1:1000), anti-phospho-S321 of mouse RIPK1 (Cell Signaling, 83613, 1:1000), anti-RIPK3 (gift from Dr. Xiaodong Wang, 1:5000), anti-phospho-S227 of RIPK3 (Abcam, ab209384, 1:1000), anti-human MLKL (GeneTex, GTX107538, 1:1000), anti-mouse MLKL (Aviva Systems Biology, OAAB10571, 1:1000), anti-phospho-S358 of MLKL (Abcam, ab187091, 1:1000), anti-phospho-S345 of mouse MLKL (Cell Signaling, 37333S, 1:1000), anti-TNFR1 (Santa Cruz, SC-8436, 1:500), anti-TRADD (BD, 610572, 1:1000), anti-TRAF2 (Cell Signaling, 4724S, 1:2000), anti-PP1γ (Prosci, 58-851, 1:1000 and Santa Cruz, SC-515943, 1:500), anti-caspase 8 (Santa Cruz, SC-6136, 1:1000, 1μg/ml for IP and SC-81656, 1:500), anti-FADD (Enzo, AAM212-E, 1:1000), anti-PARP-1 (BD, 65196E, 1:1000), anti-LDH (Abcam, ab53292, 1:2500) and anti-14-3-3 (Santa Cruz, SC-629, 1:5000). Recombinant TNF and Smac-mimetic were produced as described in ref. [17]. Rabbit anti-PPP1R3G was generated by immunizing with recombinant full-length protein fused to GST that was produced in $E.\ Coli$ (Pacific Immunology).

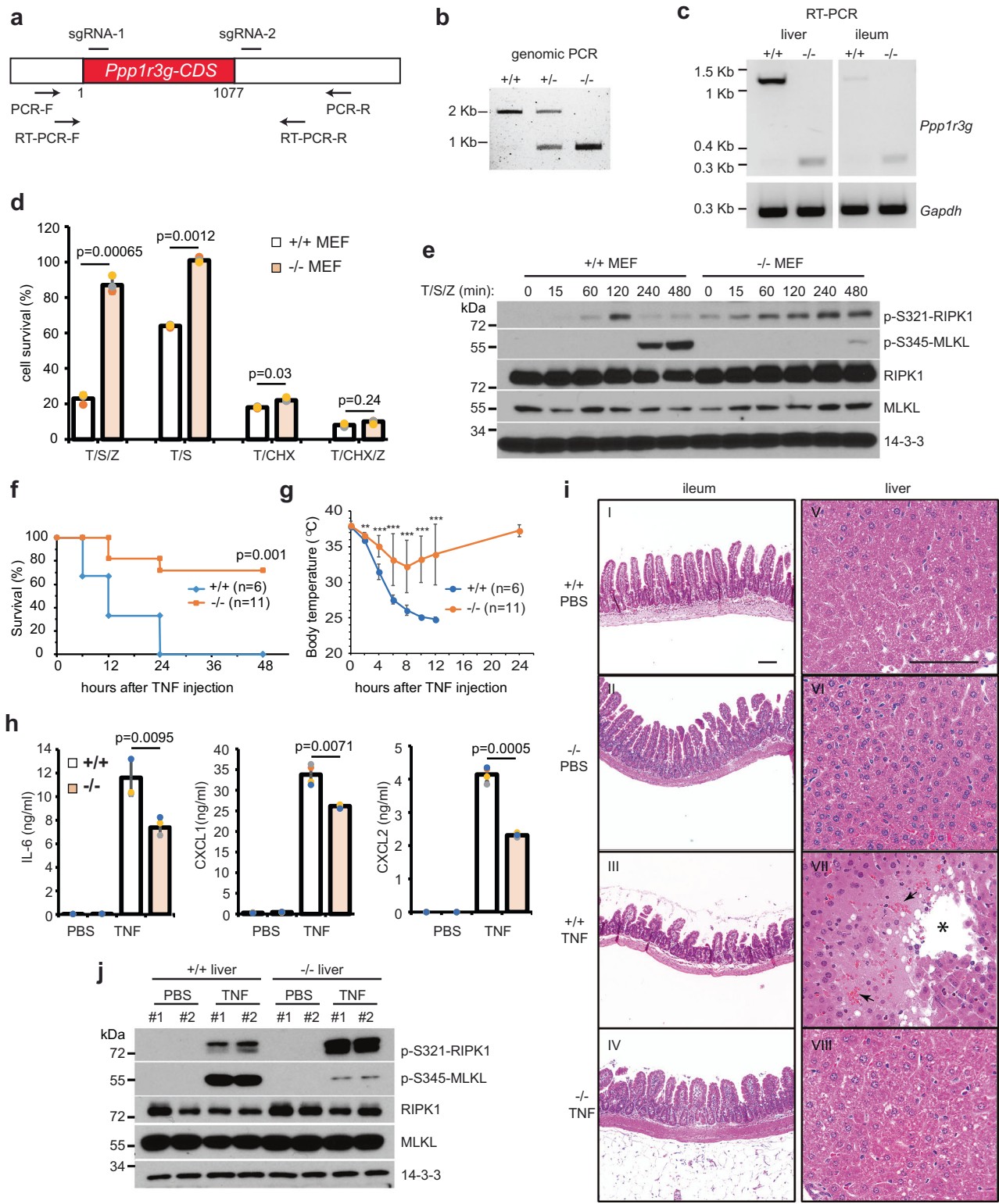

**Cell culture and stable cell lines**. HAP1 cells (Horizon Genomics) were cultured in IMDM supplemented with 10% FBS. HT-29 and MEF cells were cultured in DMEM (high glucose) supplemented with 10% FBS. (1) HAP1:RIPK3:MLKL cells. HAP1 cells were stably transfected with CMV-driven Tet Repressor (TetR, selected with 10 µg/ml Blasticidin), followed by stable expression of Dox-inducible RIPK3-DmrB fusion protein (selected with 0.5 mg/ml G418) and Dox-inducible MLKL-mCherry fusion protein (selected with 1 µg/ml puromycin). The transgene expression was induced with 50 ng/mL Dox for 24 h. (2) Knockout cell lines. CRISPR-mediated KO cell lines were generated as described before[61]. Briefly, gene-specific targeting oligos were cloned into the LentiCRISPR V2 vector (Addgene

52961), which was co-transfected with pMD2.G (Addgene 12259) and psPAX2 (Addgene 12260) into 293T cells to produce lentiviruses. Parental cells were then transduced with the viruses and single clones were selected with 1 µg/ml puromycin. RIPK1 was knocked out in PPP1R3G-KO cells to establish the PPP1R3G/RIPK1-DKO cell line. Gene knockout was confirmed by Western blotting and sequencing. The following targeting sequences were used. PPP1R3G, TGCGCTACACCTTTACCGAG and CGAGTACTGGGACAACAACG; RIPK1, TGGAAAAGGCGTGATACACA; and PP1γ, CTGGATGACAACAAGTCCGT and CTATTTGAGTCGTAGGCTGA. (3) Rescue cell lines. Tagged constructs with sense mutations at the sgRNA-targeting sites were stably expressed in the KO cells

**Fig. 7 *Ppp1r3g* $^{-/-}$ mice are protected from TNF-induced systematic inflammatory response syndrome. a** Diagram of *Ppp1r3g* knockout strategy. Positions of sgRNA targeting sites and PCR primers are shown. The majority of the *Ppp1r3g* coding sequence (CDS) is deleted. **b** Genomic PCR was performed with PCR-F and PCR-R primers. **c** RT-PCR was performed for *Ppp1r3g* and *Gapdh* from reverse transcription products of liver and ileum mRNA. **d** MEF cells from WT or knockout mice were treated with DMSO, T/S/Z, T/S, T/CHX, or T/CHX/Z for 16 h followed by CellTiter-Glo assay. Viable cells expressed as a percentage of DMSO-treated cells. Results are represented as mean ± SD of $n = 3$ biological independent samples. **e** WT or knockout MEF cells were treated with T/S/Z for various periods. Cell lysates were subjected to Western blotting with the indicated antibodies. p-S321-RIPK1 detects phosphorylated serine 321 of mouse RIPK1 and p-S345-MLKL detects phosphorylated serine 345 of mouse MLKL. **f, g** WT or knockout mice (8–14 weeks old) were injected through tail vein with mouse TNF (0.3 μg mTNF/g mouse weight). Animal survival (**f**) and body temperature (**g**) were monitored for 48 and 24 h, respectively. Kaplan−Meier survival analysis (**f**) and statistical analysis were performed with GraphPad Prism 8 (**f** and **g**). **, $P < 0.01$; ***, $P < 0.001$. **h** WT or knockout mice ($n = 4$) were injected with PBS or TNF. Serum samples were collected after 4 h and subjected to ELISA assay according to the manufacturer's protocols. **i** WT or knockout mice ($n = 3$) was injected with PBS or mTNF. Organs were harvested after 8 h from survived animals and subjected to H&E staining. Scale bar = 100 μm. * denotes a necrotic region with vacant space from hepatocyte loss and arrows denote red blood cells without vessels nearby. **j** Livers were harvested as in (**i**) and protein lysates were subjected to Western blotting with indicated antibodies.

by lentiviral transduction to generate the rescue cell lines. (4) MEF cells. E12.5 to E14.5 embryos were harvested and digested with trypsin. Fibroblast cells were plated and passed every three days. The experiments were done within five passages.

**CRISPR screen**. Whole-genome CRISPR knockout screen strategy was modified from the published protocol[62]. Human GECKO v2 library (Addgene, 1000000048) which contains 122,411 unique sgRNAs targeting 19,050 genes was used. Briefly, 40 μg library DNA, together with viral packaging vectors 10 μg pMD2.G and 30 μg psPAX2, was transfected into 4 × 15 cm plates of 293 T cells at a density of 6 × 10⁶ cells/plate. Culture media containing viruses were harvested at 48 and 72 h and viral titer was determined. Subsequently, 3 × 10⁷ viruses were transduced into 1 × 10⁸ of HAP1:RIPK3:MLKL cells in 16 × 15 cm plates at a MOI of 0.3. Two days later, the cells were split into 64 plates and selected with 1 μg/ml puromycin for 16 days. Genomic DNA was harvested from 16 plates as the pre-treatment sample. The other 48 plates were subjected to necroptosis-induction with T/S/Z for 2 days. The cells were then changed to fresh media and recovered for 5 days. Genomic DNA from the surviving cells (about 5 × 10⁶ cells) was harvested. Two rounds of PCR were performed to amplify inserted sgRNA sequences. PCR products were then subjected to next generation sequencing, SgRNAs enriched in the surviving cells represented the genes required for necroptosis. Detailed results of the screen are supplied in the Supplementary Information.

**Bioinformatics analysis**. Samples were sequenced on Illumina NextSeq 500 with read configuration as 151 bp, single end. The fastq files were subjected to quality check using fastqc (version 0.11.2, http://www.bioinformatics.babraham.ac.uk/projects/fastqc) and fastq_screen (version 0.4.4, http://www.bioinformatics.babraham.ac.uk/projects/fastq_screen), and adapters trimmed using an in-house script. The reference sgRNA sequences for human GeCKO v2.0 (A and B) were downloaded from Addgene (https://www.addgene.org/pooled-library/). The trimmed fastq files were mapped to reference sgRNA library with mismatch option as 0 using MAGeCK[63]. Further, read counts for each sgRNA were generated and median normalization was performed to adjust for library sizes. Positively and negatively selected sgRNA and genes were identified using the default parameters of MAGeCK.

**Small interfering RNA (siRNA) transfection**. For siRNA transfection, cells were plated at 2,000 cells per well in 96-well plates and 100,000 cells per well in six-well plates 24 h prior to transfection. Transfection was carried out according to the manufacturer's protocol with 5 nM siRNA/0.5 μl lipo2000 (Invitrogen) for 96 well and 50 nM siRNA/5 μl lipo2000 for 6 well. Cells were incubated in standard culture conditions for 48 h prior to treatment. Following siRNAs were used: siLuc, CGUACGCGGAAUACUUCGA; siPPP1R3G-1, GCAAGAAGCGGGUGCAGUU; and si-PPP1R3G-2, CGGCCAAGUUCCUGCAGCA. Please see Supplementary Information for other siRNA sequences.

**Cell lysates and immunoprecipitation**. Cells were scraped and washed with PBS buffer twice followed by lysing with 5 volumes of lysis buffer (50 mM Tris, pH 7.4, 137 mM NaCl, 1 mM EDTA, 1% Triton X-100, and 10% glycerol, supplemented with protease inhibitors and phosphatase inhibitors). After 30 min incubation on ice, the cells were centrifuged at 20,000 × *g* for 12 min and the supernatant was collected. Lysates (1 mg) were incubated with 20 μl anti-FLAG or anti-myc agarose beads at 4 ℃ overnight. For caspase 8 or RIPK1 IP, 1 μg anti-caspase 8 (SC-6136) or 1 μg anti-RIPK1 (BD 610459) was pre-incubated with 20 μl protein A/G beads (SC-2003) for 1 h before incubation with lysates. Beads were washed 5 times with lysis buffer and eluted with 60 μl elution buffer (0.2 M glycine, pH2.8) and immediately neutralized with 6 μl of 1 M Tris, pH 7.4. All procedures were done at 4 ℃.

**Cell survival/cell death assays**. Cell survival was measured using CellTiter-Glo Luminescent Cell Viability Assay according to the manufacturer's protocol (Promega). Cells were seeded at 2,000 cells per well in 96-well plates 24 h prior to treatment. Luminescence was measured using a BioTek Synergy 2 plate reader. For SYTOX Green staining, 10 μM SYTOX Green and 10 μg/ml DAPI were added to each well and incubated for 10 min. Annexin V-Cy3 staining was performed according to the manufacturer's protocol. Pictures were taken with a BioTek Cytation 3 machine.

**Recombinant protein purification and in vitro dephosphorylation assay**. cDNA encoding human PP1γ or PP1γ-D64N was cloned into pET-21b vector. His-tagged proteins were purified from BL21 *E.Coli* cells with Ni-NTA beads (Fisher) as described before[21]. Purified recombinant proteins were dialyzed against PBS buffer. For in vitro dephosphorylation assay, RIPK1-bound beads were incubated in PP1 buffer (50 mM Tris, pH7.4, 1 mM MnCl2, and 2 mM DTT) with 0.5 μg PP1γ, PP1γ-D64N or 400 U λPPase (NEB, P0753S) at 30 ℃ for 1 h. Mass spectrometry experiments were done following in vitro dephosphorylation and the results of the two experiments were combined.

**Liquid chromatography-mass spectrometry (LC-MS)**. The protein bands of interest were excised and destained. Samples were then digested overnight with trypsin (Pierce) following reduction and alkylation with DTT and iodoacetamide (Sigma–Aldrich). The samples then underwent solid-phase extraction cleanup with an Oasis HLB plate (Waters) and the resulting samples were injected onto an Orbitrap Fusion Lumos mass spectrometer coupled to an Ultimate 3000 RSLC-Nano liquid chromatography system. Samples were injected onto a 75 μm i.d., 75 cm long Easy-Spray column (Thermo) and eluted with a gradient from 0 to 28% buffer B over 90 min. Buffer A contained 2% (v/v) ACN and 0.1% formic acid in water, and buffer B contained 80% (v/v) ACN, 10% (v/v) trifluoroethanol, and 0.1% formic acid in water. The mass spectrometer operated in positive ion mode with a source voltage of 1.5–2.2 kV and an ion transfer tube temperature of 275 °C. MS scans were acquired at 120,000 resolution in the Orbitrap and up to 10 MS/MS spectra were obtained in the ion trap for each full spectrum acquired using higher-energy collisional dissociation (HCD) for ions with charges 2–7. Dynamic exclusion was set for 25 s after an ion was selected for fragmentation. Raw MS data files were analyzed using Proteome Discoverer v2.2 or 2.4 (Thermo), with peptide identification performed using Sequest HT searching against the human protein database from UniProt. Fragment and precursor tolerances of 10 ppm and 0.6 Da were specified, and three missed cleavages were allowed. Carbamidomethylation of Cys was set as a fixed modification, with oxidation of Met set as a variable modification and phosphorylation of Ser, Thr, and Tyr also set as a variable modification for phosphorylation searches. The false-discovery rate (FDR) cutoff was 1% for all peptides. Please see Supplementary Information for detailed mass spectrometry results.

**Constructs**. cDNAs were PCR cloned from reverse transcription products from HT-29 cells. The primers were designed according to the following reference sequences, RIPK3 (NM_006871.3), RIPK1 (NM_003804.3), MLKL (NM_152649), PPP1R3G (NM_001145115), PP1γ (NM_002710), and mouse PPP1R3G (NM_029628). All mammalian expression constructs were cloned into CMV-driven pcDNA3-based vectors (Invitrogen). Dox-inducible expression was based on the pCDNA 4/TO vector (Invitrogen). All point mutations were generated by site-directed mutagenesis and verified by sequencing.

**qPCR**. Total RNA was isolated with Direct-zol™ RNA Kits (Zymo Research) and digested with DNase I. cDNA was synthesized using iScript cDNA synthesis kit (BioRad, 1708891). Gene expression was assessed using standard qPCR approaches with iTaq™ universal SYBR® Green supermix (172-5124). Analysis was performed on CFX Connect™ Real-Time PCR Detection System (BioRad). The $2^{-\Delta\Delta Ct}$ method was used to analyze the relative change in gene expression normalized to actin. The following primers were used for qPCR: *Actin*, AACTCCATCATGAAGTGTGACG,

and GATCCACATCTGCTGGAAGG; *PPP1R3G*, TCTTCTCTGTGCTGCTTCCA, and CTTTGCCATCTGGTCTCTGC.

**Mouse work**. Straight knockout mice were generated using CRISPR-Cas9 technology as described before[64]. The two sgRNA sequences were CGGAGACCCU-CAAUCCGCGG and AGAGGUUGACCAGCUAGCAA. The following primers were used for genotyping: PCR-F, CAAAGCTATCGGCTAGCCTACGAC, and PCR-R, ACATGTTCACGCGGCCTTTACTTT; RT-PCR-F, CCGCGGTTCCCGC AGGACGAAACC, and RT-PCR-R, AGAACACTGCCAGAGCTTGCCGTG. F1 mice were backcrossed to C57BL/6 background. For SIRS model, 8–14 weeks old mice were injected through tail vein with mouse TNF (0.3 µg TNF/g body weight). The temperature was monitored for 24 h with a homeothermic monitoring system (Harvard Apparatus) and survival was monitored for 48 h. Blood samples were collected 4 h after injection by retro-orbital method and cytokine levels were determined by ELISA according to the manufacturer's protocols. Tissues were harvested 8 h after injection and standard H&E staining was performed by the UT Southwestern Histo Pathology Core.

**Statistical analysis and reproducibility**. Statistical analyses were performed with Excel and GraphPad Prism 8. CellTiter-Glo results are presented as mean $+/-$ SD of $n = 3$ biological independent samples. Two-tailed Student's t-test is performed to determine statistical significance. $*P < 0.05$; $**P < 0.01$; $***P < 0.001$. Western blotting data are representative of at least three independent experiments with similar results. See Supplementary Fig. 3 for uncropped blots.

**Reporting summary**. Further information on research design is available in the Nature Research Reporting Summary linked to this article.

## Data availability

CRISPR screening data generated in this study have been deposited to GEO with an identification number GSE176422. All other data generated during the current study are included in this published article and its Supplementary Information files. Source data are provided with this paper.

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

## Acknowledgements

We would like to thank Dr. Zhijian "James" Chen, Dr. Youtong Wu, Dr. Chee-Kwee Ea for helping with the CRISPR screen as well as reagents and critical discussions. Thanks to Dr. Eric Olson and John McAnally for generating the knockout mice. Thanks to Dr. Pascal Meier for phospho-S320-RIPK1 antibody. Thanks to Hong Yu, Dr. Mohammad Goodarzi, and Dr. Andrew Lemoff at the UT Southwestern Proteomics Core, Dr. Bret Evers, and John Shelton at the UT Southwestern Histo Pathology Core for excellent technical assistance. Thanks to Dr. Rhonda Bassel-Duby and Dr. Thomas Carroll for the critical reading of the manuscript. This work is supported by grants from the Welch Foundation (I1827) and NIGMS (R01, GM120502) to Z.W. Z.W. is the Virginia Murchison Linthicum Scholar in Medical Research, and Cancer Prevention and Research Institute of Texas Scholar (R1222).

## Author contributions

J.D., Y.X., and Z.W. conceived and designed the study. J.D., Y.X., H.L., S. L., and Z.W. performed experiments and analysis. A.K. and C.X. conducted bioinformatics analysis.

## Competing interests

The authors declare no competing interests.
