## [Peer Review File · Nature Communications]

RIPK1 dephosphorylation and kinase activation by PPP1R3G/PP1 γ promote apoptosis and necroptosisREVIEWER COMMENTS

Reviewer #1 (Remarks to the Author):

The manuscript by Wang and colleagues identifies a new regulatory node in the activation of apoptosis and necroptosis by RIPK1 kinase. It is well established that RIPK1 undergoes multiple inhibitory phosphorylations that necessitate inhibition of various kinases (such as TAK1, MK2, IKKs) to promote cell death. However, how these phosphorylations are countered by endogenous phosphatases is unknown. The authors demonstrate that PPP1R3G/PP1g plays a key role in de-phosphorylation of RIPK1, which promotes Complex II formation and cell death. Overall, I am convinced based on the data that PPP1R3G/PP1g can de-phosphorylate RIPK1 on multiple inhibitory sites. The experiments in the paper are of high quality and, in principle, the message is timely and is of general interest. However, I am not convinced that the authors identified the mechanism of this regulation correctly.

1. In Fig. 3 the authors examined PPP1R3G binding partners. Curiously, a number of metabolic enzymes were identified, but not pursued. This was an oversight because the contribution of metabolism to necroptosis has been previously suggested. Furthermore, a different isoform of PYGB, PYGL, was found to be a component of Complex II (PMID: 19498109). There is insufficient direct evidence that PP1g is the correct binding partner for PPP1R3G in this regulation. While RAQA mutation inhibits PP1g binding, it is not shown whether it affects other interactions. There is also no data using PP1g loss-of-function to directly demonstrate its role in Complex 2/cell death.

2. In Fig. 4, the authors interpret the data to establish that PPP1R3G/PP1g is recruited to Complex I. However, it is well established that Complex I forms within minutes after stimulation and generally disappears within an hour. I don't think experiments in Fig. 4B, performed at 4 hr time point, reflect PPP1R3G/PP1g recruitment into Complex I. To verify this conclusion, the authors need to either pull down TNF or TNFR1 after TNF alone treatment for a short period of time and examine recruitment to Complex I. The authors most likely see Complex II in Fig. 4B.

3. It is confusing which paradigms of cell death are affected by the loss of PPP1R3G/PP1g.

a) Why is necroptosis induced by TNF/CHX/zVAD not affected by KO in Fig. 2H?

b) I am not convinced by data in Fig. 5B. For example, 5z-7 is well established to be a potent promoter of cell death at 100-200 nM in HT29 cells. Why is 10 uM concentration required? It looks to me that KO cells are still quite resistant to all of the IKK and TAK1 inhibitors in contrast to the conclusion by the authors. The authors need to compare side by side KO and WT cells with these same treatments in Fig. 5B.

c) The experiments are limited to HT29 cells. While these cells are widely used, there are many other cellular systems that are equally widely used. Furthermore, it is unclear whether role of PPP1R3G/PP1g is limited to paradigms involving IAP inhibition or not. For example, the authors can use FADD-deficient Jurkat or L929 cells that are directly sensitive to TNF alone without SMACi or MEFs that are very sensitive to TNF/zVAD without SMACi. In particular, MEF cells could be expected to become highly sensitive to TNF alone upon KO of PPP1R3G if the authors' model is correct.

d) In Fig. 5F, the authors show a well-accepted model suggesting that RIPK1 undergoes inhibitory phosphorylations in Complex I. The conundrum is that these events require RIPK1 ubiquitination. How would this model work in the presence of IAP antagonist, blocking RIPK1 ubiquitination? However, Ser320 phosphorylation by MK2 is not actually happening in Complex I, it affects cytosolic pool of RIPK1 (PMID: 28506461). I suspect that PPP1R3G also affects the cytosolic pool of RIPK1 and not Complex I.

Minor points:

- 1) I will need to re-examine the paper since supplementary file is corrupted and is un-readable.
- 2) The authors need to include the full results of the CRISPR screen in the supplement.

Reviewer #2 (Remarks to the Author):

In this study, Du et al. aim to identify the mechanism of RIPK1 dephosphorylation in necroptosis. They first conducted a genome-wide CRISPR knockout screen using the library GeCKO V2 in modified HAP1 cells (HAP1:RIPK3:MLKL) treated with TNF, Smac-mimetic, and Z-VAD-FMK (T/S/Z) to induce necroptosis. They sequenced these cells against an early time point and identified genes essential for necroptosis, including PPP1R3G. They then conducted a secondary siRNA screen to target the top 60 hits from the primary screen in HAP1:RIPK3:MLKL cells. They found that loss of PPP1R3G inhibited necroptosis, and confirmed this result in unmodified HT-29 cells. Next, they conducted a series of immunoprecipitation experiments to elucidate the mechanism in question and found that the PPP1R3G/PP1 γ holoenzyme dephosphorylates sites on RIPK1 to activate both necroptosis (T/S/Z treatment) and apoptosis (T/S treatment). Finally, they studied this mechanism in in vivo mouse models, and found Ppp1r3g^{-/-} mice were resistant to TNF-induced SIRS, confirming the essentiality of PPP1R3G in cell death. The authors hope that the essential interaction of PPP1R3G/PP1 γ may serve as an important therapeutic target pathway for inflammatory and degenerative diseases caused by RIPK1 mutations.

We were requested to comment on the CRISPR screens presented in this paper, therefore our major points will be focused on Figures 1 and 2. Below we provide suggested additions to strengthen the authors' conclusions. Overall, we believe the manuscript would have a greater impact if screening details were made more transparent and all data made available.

Major points:

- 1) Du et al. conducted a genome-wide CRISPR screen, but details and results are sparse. Most notably, the raw screening data (e.g. read counts of individual guides) are not included in the results, nor is there

an output of gene-level conclusions (i.e. the information summarized in Figure 1F). Additionally, the authors should present how negative control guides behaved, in order to provide insight on the levels of the noise in the screen. Such transparency would also allow the manuscript to have greater impact in the scientific community at large by making the data usable for future analysis by others.

2) The authors should include data regarding the quality of their CRISPR and siRNA screens. Please include information on the number of replicates and a demonstration of replicate reproducibility.

3) Fig. 2 shows the siRNA screen results for only one control and PPP1R3G, but the manuscript mentions that the screen contained siRNAs targeting the top 60 hits from the primary screen and that there were “many other positive hits” which may have efficiently inhibited necroptosis, without further comment on this screen or its results. Again, screening data should be provided. What were the other hits? How many of these hits were validated? How many siRNAs were present for each of these and what were their sequences? What was the validation rate? Were other negative controls used? It is also concerning that the only control used appears to have been one siRNA targeting luciferase; please see this commentary from Bill Kaelin on why this is insufficient as a negative control (PMID: 22837515). Neither the manuscript description nor the Experimental Procedures section for siRNA transfection is detailed enough to understand or reproduce this screen, therefore these results cannot be supported as is without further transparency.

4) Fig. 1A shows a faint RIPK3-DmrB band in the no Dox condition. Some rationale should be included regarding why the Dox leakiness is not an issue. Similarly, Fig. 1B-C would be strengthened by including with and without Dox conditions to support those conclusions.

5) The authors rely on T/S/Z treatment combinations throughout their manuscript as a reliable method for inducing necroptosis in their cell lines, without an explicit comment on this methodology. Are there any concerns about off-target effects of T/S/Z treatment? Were dosage titrations done on the cell lines prior to screening or is there a reference from which dosages were chosen?

Minor points:

1) The authors should comment on the choice of the GeCKO V2 library. Specifically, the correlation and usage of Set A and B, and what is known about how the guides interact with endogenous RIPK, MLKL – that is, which guides will target the endogenous loci but not the introduced cDNAs. This is particularly important to know due to the highly-engineered nature of the cell line used.

2) There is very little comment on the choice of HAP1 cells for the genome-wide CRISPR screen. Is there a rationale behind why these are a good model for necroptosis or quantification supporting that they are “highly sensitive to necroptosis”? Does the cell line’s haploid status impact the results and if so, how frequently were they sorted during passaging to maintain this characteristic?

3) Fig. 1E could be more valuable if there was information about which genes contributed to the results (relating to Major point 1). Perhaps consider linking information about the hits shown in Fig. 1F to the appropriate pathway in Fig. 1E to show the impact of the different hits.

4) It would be beneficial to include more data in the CRISPR Screen Experimental Procedures section regarding any tangible experimental numbers in the treatment arm. For example, there is mention of cell numbers at the beginning of the CRISPR screen, but how many cells recovered in the 5 days after T/S/Z treatment? How much genomic DNA was sequenced at the end of the screen, i.e. what was the representation?

Reviewer #3 (Remarks to the Author):

I was asked specifically to review the mass spectrometry portion of this manuscript. Although the authors mention mass spectrometry in the body of the paper, they really do not show any of the data in the manuscript, nor did they provide complete results. There is no list of identified proteins/peptides; these details should be provided in a supplementary table, at least. As it stands, there is no way to evaluate the mass spectrometry results.

The mass spectrometry section in Materials and Methods needs much work, as most essential details have been omitted. Not enough details are provided to be able to ascertain exactly what was done. No reduction/alkylation protocol was given. The samples were analyzed on a Lumos, but no details were provided for the analysis itself (LC gradient, mass spec acquisition parameters). Further, absolutely no details were provided for the data analysis section other than the search engine and the protein database. Fixed modifications? Variable modifications? Mass error tolerances? The authors should, at the very least, provide references for procedures that were used if space is an issue.

one key correction: it is Liquid chromatography-mass spectrometry (LC-MS).

The data upload in MassIVE contains only minimum data. Only the raw files were provided, along with a .xml file that could not be opened.

Reviewer #4 (Remarks to the Author):

The authors investigate the dephosphorylation of RIPK1 that leads to its activation to promote apoptosis and necroptosis. They start out with a Crispr/Cas9 screen in which they identify PPP1R3G as a gene required for necroptosis, and follow this up with a detailed mechanistic study using molecular cell biology, biochemistry, mass spectrometry, and mouse biology approaches. This is a thorough study which I enjoyed reading. I think that this is timely and well carried out. The manuscript is well written but requires a bit of English polishing (particularly regarding the use of articles). I have some concerns that I think should be addressed before acceptance:

- 1) P. 2 line 21: "The PP1 holoenzyme is an obligatory heteromer composed of a PP1 catalytic subunit (PP1c) and one or two regulatory subunits, also referred to as PP1-interacting proteins (PIPs)." The PIPs have been renamed, and I suggest to use maybe the old version but also mention the new name of them: regulatory interactors of protein phosphatase one (RIPPOs) (PMID: 30115685; PMID: 32956763)
- 2) The following sentence and later in the manuscript p.4 line 23: The authors write "by regulatory subunits or PIPs". PIPs are regulatory subunits, this distinction is unclear to me (and in disagreement with their previous sentence).
- 3) Smac-mimetic: I could not find which Smac mimetic the authors refer to, also not in the methods. Maybe I overlooked it.
- 4) p.4 line 23/24: "PP1 regulatory subunits or PIPs often contain a PP1c binding motif and a substrate binding motif." Aside from the distinction between PIPs and regulatory subunits, which is unclear to me, here the "substrate binding motif" is unclear. Do the authors have a citation for this or do they confuse this with PP2A B-subunits? I would delete this sentence as well as the "and" at the beginning of the next sentence, and just leave the rest of the next sentence.
- 5) Fig. 4 is split between 2 chapters, I recommend to merge the two chapters for a clearer structure.
- 6) P. 6 line 12: It should be Fig. 4E, not 4F
- 7) The manuscript contains many Western blots, but I am not sure how often the experiments were repeated. That should be clearly written in the captions. They were not quantified. While, at large, that might not be necessary in every case, in Fig. 6C that should be done as the difference between lanes 3 and 9 is not easily visible and the conclusion (p.8 line 1) is therefore in this case not convincing. Sometimes the blots are overexposed, like in Fig 2J (pS166), Fig 4B (FLAG-PPP1R3G, PP1gamma) or 4C (RIPK1 and PP1gamma), Fig 6C (MLKL). Why was 14-3-3 used in some cases for loading control, not a more commonly used loading control like actin, tubulin or GAPDH?

8) Fig. 5B and C shows effects of compound treatment in PPP1R3G-KO cells only, based on the assumption that PP1 activity will not be required to activate cell death if the phosphorylation is already inhibited. The comparison to WT is lacking, it should be the same according to their theory. Particularly because in Fig. 5D, where the experiment is done based on the previous findings, then the authors use WT cells, which is then not comparable to the previous experiments.

9) Fig. 5F and 3G mass spectrometry data: The full data needs to be made available via supplementary table.

10) The revision should at least include the source data for the Western blots, if not all source data.

11) The in vitro experiments are done only with catalytic subunit and not with the holoenzyme. I think that this is acceptable because the isolation of the holoenzyme is not simple, the reaction was controlled by lambda-phosphatase, and PP1gamma was used. Direct dephosphorylation was shown to be possible.

12) P. 7 line 6: "S320 phosphorylation of RIPK1 reached high level at 15 minutes after T/S/Z treatment and decreased dramatically at 4 hours, at which time S166 phosphorylation of RIPK1 was highest (Fig. 5D), confirming previously reported inhibitory role of S320 phosphorylation." I disagree, this only shows a sequential phosphorylation/dephosphorylation, it is not an activity assay. It should be rephrased.

13) In the last chapter on the knockout mouse, more information on the mouse model and the disease model is necessary to understand this fully. How does the SIRS model work? For example, the sentences from the discussion could be moved here: "Ppp1r3g-deficient mice are viable and have less glycogen deposition in liver. They also have reduced body weight and fat composition upon high-fat diet feeding⁵³." At the moment it sounds like they were the first ones to create the knockout model. Also "SIRS is an excessive defense response of the body to insults, including infection, trauma, acute inflammation, and ischemia. TNF-induced SIRS is a mouse model of sterile sepsis⁴³, widely used in recent times to evaluate the in vivo function of necroptosis pathway components." can be moved into the mouse results chapter from the discussion.

14) Fig. 7: Why do the authors look at ileum and liver? In Fig. 7c ileum should be shown in addition to liver because both tissues were looked at with H&E staining, and the protein levels should also be compared between wt and knockout tissues.

15) Fig. 2C caption: why does PPP1R3G run at 55kDa? Is this known or how do the authors know what they are looking at?

Overall, I recommend acceptance of this manuscript once the comments have been addressed.

Here are our point-by-point response to the reviewers' comments which are reiterated in italics.

Reviewer #1:

1. In Fig. 3 the authors examined PPP1R3G binding partners. Curiously, a number of metabolic enzymes were identified, but not pursued. This was an oversight because the contribution of metabolism to necroptosis has been previously suggested. Furthermore, a different isoform of PYGB, PYGL, was found to be a component of Complex II (PMID: 19498109). There is insufficient direct evidence that PP1g is the correct binding partner for PPP1R3G in this regulation. While RAQA mutation inhibits PP1g binding, it is not shown whether it affects other interactions. There is also no data using PP1g loss-of-function to directly demonstrate its role in Complex 2/cell death.

The reviewer raised a very good point. We focused on PP1 γ mainly because PPP1R3G was a known phosphatase subunit. In revised Fig. 3B, we showed that the RAQA mutant did interact with PYGB as strongly as the WT PPP1R3G protein, suggesting that failure to interact with PP1 γ , not PYGB is responsible for RAQA mutant deficiency. And we also showed that silencing PP1 γ inhibited necroptosis (Fig. 3C and 3D). These results demonstrate the essential function of both PPP1R3G and PP1 γ for necroptosis.

2. In Fig. 4, the authors interpret the data to establish that PPP1R3G/PP1g is recruited to Complex I. However, it is well established that Complex I forms within minutes after stimulation and generally disappears within an hour. I don't think experiments in Fig. 4B, performed at 4 hr time point, reflect PPP1R3G/PP1g recruitment into Complex I. To verify this conclusion, the authors need to either pull down TNF or TNFR1 after TNF alone treatment for a short period of time and examine recruitment to Complex I. The authors most likely see Complex II in Fig. 4B.

That is a good point. PPP1R3G/PP1 γ possibly interacts with TRAF2/RIPK1 in multiple protein complexes, including complex I. We performed FLAG-TNF IP experiment and PPP1R3G/PP1 γ was detected in the TNF/TNFR1/RIPK1/TRAF2-containing complex I, after T/S/Z treatment for 30' (Fig. 4C). Importantly, phospho-S166 of RIPK1 was only detected in complex I after T/S/Z induction in WT cells, but not in KO cells, suggesting that RIPK1 activation in complex I is inhibited without PPP1R3G.

3. It is confusing which paradigms of cell death are affected by the loss of PPP1R3G/PP1g.

a) Why is necroptosis induced by TNF/CHX/zVAD not affected by KO in Fig. 2H?

Thanks to the reviewer for raising this very important point, which we did not pay sufficient amount of attention in the previous draft. Our data suggest that PPP1R3G/PP1 γ regulates RIPK1 kinase activation. It is easy to understand in the case of T/CHX-induced apoptosis, which is independent of RIPK1 (PMID: 18485876, reference 14). Regarding necroptosis, an important recent development is the recognition of two different modes of necroptosis by Dr. Junying Yuan's group (PMID: 29891719, reference 15). They propose that there are two types of necroptosis. Type I is induced by T/5z-7/Z or T/S/Z, which requires kinase activation of RIPK1 in complex I. In contrast, type II is induced by T/CHX/Z, which does not require RIPK1 activation in complex I. Instead, it only requires RIPK1 kinase activity in necrosome to phosphorylate RIPK3. In that report, deficiency in LRRK2 and c-Cbl, a kinase and an E3 ligase respectively, inhibits T/S/Z-induced type I necroptosis, but not T/CHX/Z-induced type II necroptosis, which is similar to PPP1R3G knockout. Notably, like PPP1R3G/PP1 γ , c-Cbl interacts with complex I and is required for RIPK1 activation in complex I. Therefore, we believe that PPP1R3G/PP1 γ is essential for RIPK1-dependent apoptosis and type I necroptosis through activating RIPK1 kinase in complex I. We have commented on the role of PPP1R3G in these two pathways in the revised manuscript.

b) I am not convinced by data in Fig. 5B. For example, 5z-7 is well established to be a potent promoter of cell death at 100-200 nM in HT29 cells. Why is 10 μ M concentration required? It looks to me that KO cells are still quite resistant to all of the IKK and TAK1 inhibitors in contrast to the conclusion by the authors. The authors need to compare side by side KO and WT cells with these same treatments in Fig. 5B.

We have replaced 5Z-7 with a p38 inhibitor in the revised Fig. 5B. These inhibitors did enhance cell death in HT-29 cells, but not to the extent as in PPP1R3G-KO cells.

c) *The experiments are limited to HT29 cells. While these cells are widely used, there are many other cellular systems that are equally widely used. Furthermore, it is unclear whether role of PPP1R3G/PP1γ is limited to paradigms involving IAP inhibition or not. For example, the authors can use FADD-deficient Jurkat or L929 cells that are directly sensitive to TNF alone without SMACi or MEFs that are very sensitive to TNF/zVAD without SMACi. In particular, MEF cells could be expected to become highly sensitive to TNF alone upon KO of PPP1R3G if the authors' model is correct.*

We have shown in the manuscript that PPP1R3G/PP1γ is required for RIPK1 activation-dependent cell death in different cell lines, including modified HAP1, HT-29 and MEF cells. Per the reviewer's request, we examined T/Z-induced cell death in MEF cells. PPP1R3G-KO did not inhibit T/Z-induced necroptosis in MEF cells, likely because this type of cell death is similar to T/CHX/Z-induced type II necroptosis. Our model also predicts that PPP1R3G-KO cells would have higher level of RIPK1 inhibitory phosphorylations. As a consequence, these cells are more resistant to RIPK1-dependent cell death.

d) *In Fig. 5F, the authors show a well-accepted model suggesting that RIPK1 undergoes inhibitory phosphorylations in Complex I. The conundrum is that these events require RIPK1 ubiquitination. How would this model work in the presence of IAP antagonist, blocking RIPK1 ubiquitination?*

RIPK1 ubiquitination is a complex process. Many E3 ligases, including cIAPs, LUBAC and recently reported c-Cbl, promote polyubiquitination of RIPK1. As reported by de Almargo et al (PMID: 26111062), in the presence of TNF, IAP antagonist BV-6 and Z-VAD-FMK (T/B/Z), RIPK1 is polyubiquitinated by LUBAC during necroptosis. Furthermore, in cIAP1^{-/-}cIAP2^{-/-} MEF cells, T/B/Z induced comparable level of RIPK1 polyubiquitin as in WT MEF cells. These results suggest that RIPK1 will still undergo polyubiquitination catalyzed by LUBAC in the absence of IAP proteins. More importantly, it has been shown that another E3 ligase, c-Cbl is recruited to complex I to promote K63-polyubiquitination of RIPK1, independent of c-IAPs (PMID: 29891719, reference 15). Notably, c-Cbl deficiency inhibits RIPK1-dependent apoptosis induced by T/5z-7 or T/S, but not apoptosis-induced by T/CHX. Similar to PPP1R3G-KO, c-Cbl deficiency also inhibits necroptosis induced by T/5Z-7/Z or T/S/Z, but not necroptosis induced by T/CHX/Z.

However, Ser320 phosphorylation by MK2 is not actually happening in Complex I, it affects cytosolic pool of RIPK1 (PMID: 28506461). I suspect that PPP1R3G also affects the cytosolic pool of RIPK1 and not Complex I.

S320 phosphorylation happens in both complex I and cytosol (PMID: 28506461, reference 35). The reviewer raised a very good point. We believe that **PPP1R3G/PP1γ** could associate with RIPK1 both in cytosol and complex I to dephosphorylate RIPK1. Basal association was detected without treatment (lane 3, Fig. 4B) and that association increased drastically after T/S/Z treatment (lane 4, Fig. 4B). We have commented on this point in the results section and the discussion section.

Minor points:

- 1) I will need to re-examine the paper since supplementary file is corrupted and is un-readable.*
- 2) The authors need to induce the full results of the CRISPR screen in the supplement.*

Detailed screening results are provided in Supplementary Table 1-5.

Reviewer #2:

- 1) Du et al. conducted a genome-wide CRISPR screen, but details and results are sparse. Most notably, the raw screening data (e.g. read counts of individual guides) are not included in the results, nor is there an output of gene-level conclusions*

(i.e. the information summarized in Figure 1F). Additionally, the authors should present how negative control guides behaved, in order to provide insight on the levels of the noise in the screen. Such transparency would also allow the manuscript to have greater impact in the scientific community at large by making the data usable for future analysis by others.

Thanks to the reviewer's suggestions, we have submitted two files to GEO. (1) FASTQ files; and (2) Median normalized counts of sgRNA generated by MAGeCK (reference 63). Please retrieve the files using the following link, Go to <https://www.ncbi.nlm.nih.gov/geo/query/acc.cgi?acc=GSE176422>
Enter token opwvimaibzwpstet into the box.

In addition, the following files are provided in the supplementary information. (1) Table 1, Screening gene summary. Genes are ranked according to the positive p-value. (2) Table 2, Normalized counts. Median normalized counts of sgRNA were generated by MAGeCK. Same file as in GEO. (3) Table 3, Non-targeting control guides. Non-targeting guides are critical in evaluating the noise in the screen.

2) The authors should include data regarding the quality of their CRISPR and siRNA screens. Please include information on the number of replicates and a demonstration of replicate reproducibility.

The GeCKOV2 library contains two sets of sgRNA for each gene, including 3 sgRNAs in library A and 3 sgRNAs in library B. To show the correlation between Library A and Library B, we have attached the principal component analysis plot performed on the most significant guide for each gene in GeCKO V2 library A and B (Supplementary Figure 1). The most significant guides were selected based on lowest p-value significance for read count difference of each sgRNA in two conditions. We have also shown the spearman correlation p-value significance between the samples. P-value significance of 0.0001 was found between Control_A and Control_B. P-value significance of 0.0027 was found between T/S/Z_A and T/S/Z_B.

3) Fig. 2 shows the siRNA screen results for only one control and PPP1R3G, but the manuscript mentions that the screen contained siRNAs targeting the top 60 hits from the primary screen and that there were "many other positive hits" which may have efficiently inhibited necroptosis, without further comment on this screen or its results. Again, screening data should be provided. What were the other hits? How many of these hits were validated? How many siRNAs were present for each of these and what were their sequences? What was the validation rate? Were other negative controls used? It is also concerning that the only control used appears to have been one siRNA targeting luciferase; please see this commentary from Bill Kaelin on why this is insufficient as a negative control (PMID: 22837515). Neither the manuscript description nor the Experimental Procedures section for siRNA transfection is detailed enough to understand or reproduce this screen, therefore these results cannot be supported as is without further transparency.

The siRNA sequences and their effect are listed in the Supplementary Table 5 and Supplementary Figure 2. At least two different siRNAs were used for each gene, except that a few genes were targeted by four siRNAs. Two of negative siRNAs were used and transfection protocol was updated in the experimental procedure section.

4) Fig. 1A shows a faint RIPK3-DmrB band in the no Dox condition. Some rationale should be included regarding why the Dox leakiness is not an issue. Similarly, Fig. 1B-C would be strengthened by including with and without Dox conditions to support those conclusions.

Samples without Dox induction were included in the revised Fig. 1B and 1C. Without Dox, the leaked expression does not promote cell death.

5) The authors rely on T/S/Z treatment combinations throughout their manuscript as a reliable method for inducing necroptosis in their cell lines, without an explicit comment on this methodology. Are there any concerns about off-target effects of T/S/Z treatment? Were dosage titrations done on the cell lines prior to screening or is there a reference from which dosages were chosen?

T/S/Z treatment was described in the experimental procedure according to reference 16. Titration was done and the condition induced maximum cell death in modified HAP1 cells. We were not concerned about off-target effect since a RIPK1 inhibitor necrostatin (Nec-1) completely blocked T/S/Z-induced necroptosis in these cells.

Minor points:

1) The authors should comment on the choice of the GeCKO V2 library. Specifically, the correlation and usage of Set A and B, and what is known about how the guides interact with endogenous RIPK, MLKL – that is, which guides will target the endogenous loci but not the introduced cDNAs. This is particularly important to know due to the highly-engineered nature of the cell line used.

As discussed in point 2, the spearman correlation p value for set A and set B was 0.0001 for the control sample and 0.0027 for T/S/Z-treated sample (Supplementary Figure 1). Per the reviewer's request, we also provided the sequence of the 12 sgRNAs in the library for RIPK3 and MLKL, as well as their relationship to the transgenes (Supplementary Table 4). In the end, 4 out of 5 sgRNA targeting transgene RIPK3-DmrB and 5 out of 5 sgRNA targeting transgene MLKL-mCherry were highly enriched in the survived cells.

2) There is very little comment on the choice of HAP1 cells for the genome-wide CRISPR screen. Is there a rationale behind why these are a good model for necroptosis or quantification supporting that they are “highly sensitive to necroptosis”? Does the cell line's haploid status impact the results and if so, how frequently were they sorted during passaging to maintain this characteristic?

We chose HAP1 cells because of their haploid genome, which in theory sensitizes the screen since only one allele needs to be disrupted by CRISPR-cas9. The engineered cells were almost 100% dead with T/S/Z treatment, thus “highly sensitive to necroptosis”. We used freshly-generated cells for screening and we did not sort the cells to maintain its haploid status, but we did sort cells for mCherry signal to maintain high MLKL-mCherry expression in following studies.

3) Fig. 1E could be more valuable if there was information about which genes contributed to the results (relating to Major point 1). Perhaps consider linking information about the hits shown in Fig. 1F to the appropriate pathway in Fig. 1E to show the impact of the different hits.

All top 10 genes except PPP1R3G are linked to cell death signaling in literature. We did mention that “Known necroptosis players MLKL, RIPK1, RIPK3 and TNFR1 (gene *TNFRSF1A*) were among the top 5 positive hits, validating the effectiveness of the screen”.

4) It would be beneficial to include more data in the CRISPR Screen Experimental Procedures section regarding any tangible experimental numbers in the treatment arm. For example, there is mention of cell numbers at the beginning of the CRISPR screen, but how many cells recovered in the 5 days after T/S/Z treatment? How much genomic DNA was sequenced at the end of the screen, i.e. what was the representation?

The detailed protocol was described in reference 61. About 5×10^6 cells were harvested at the end. We did not sequence genomic DNA directly. Instead, two rounds of PCR were conducted, and gel purified PCR products were sent for sequencing. Upon sequencing, 119,160 of total 119,462 (99.7%) sgRNA were detected in the control group, suggesting almost complete coverage of the library in the starting cell population (Supplementary Table 2).

Reviewer #3

I was asked specifically to review the mass spectrometry portion of this manuscript. Although the authors mention mass spectrometry in the body of the paper, they really do not show any of the data in the manuscript, nor did they provide complete results. There is no list of identified proteins/peptides; these details should be provided in a supplementary table, at least. As it stands, there is no way to evaluate the mass spectrometry results.

Identified proteins and peptides were listed in the Supplementary Table 6 and 7.

The mass spectrometry section in Materials and Methods needs much work, as most essential details have been omitted.

Not enough details are provided to be able to ascertain exactly what was done. No reduction/alkylation protocol was given. The samples were analyzed on a Lumos, but no details were provided for the analysis itself (LC gradient, mass spec acquisition parameters). Further, absolutely no details were provided for the data analysis section other than the search engine and the protein database. Fixed modifications? Variable modifications? Mass error tolerances? The authors should, at the very least, provide references for procedures that were used if space is an issue.

The Experimental Procedures section about LC-MS has been updated to cover the details.

one key correction: it is Liquid chromatography-mass spectrometry (LC-MS).

Corrected. Thanks.

The data upload in MassIVE contains only minimum data. Only the raw files were provided, along with a .xml file that could not be opened.

New uploads have been submitted. Please follow the link for more information:

<http://massive.ucsd.edu/ProteoSAFe/status.jsp?task=26f1c9364c4d4a11b2108c3f82b95584>

Password: Zhigao

Reviewer #4

1) P. 2 line 21: "The PP1 holoenzyme is an obligatory heteromer composed of a PP1 catalytic subunit (PP1c) and one or two regulatory subunits, also referred to as PP1-interacting proteins (PIPs)." The PIPs have been renamed, and I suggest to use maybe the old version but also mention the new name of them: regulatory interactors of protein phosphatase one (RIPPOs) (PMID: 30115685; PMID: 32956763)

This part has been rewritten as requested.

2) The following sentence and later in the manuscript p.4 line 23: The authors write "by regulatory subunits or PIPs". PIPs are regulatory subunits, this distinction is unclear to me (and in disagreement with their previous sentence).

This part has been rewritten.

3) Smac-mimetic: I could not find which Smac mimetic the authors refer to, also not in the methods. Maybe I overlooked it.

It has been updated in the Experimental Procedures.

4) p.4 line 23/24: "PP1 regulatory subunits or PIPs often contain a PP1c binding motif and a substrate binding motif." Aside from the distinction between PIPs and regulatory subunits, which is unclear to me, here the "substrate binding motif" is unclear. Do the authors have a citation for this or do they confuse this with PP2A B-subunits? I would delete this sentence as well as the "and" at the beginning of the next sentence, and just leave the rest of the next sentence.

This part has been rewritten.

5) Fig. 4 is split between 2 chapters, I recommend to merge the two chapters for a clearer structure.

This part has been rewritten.

6) P. 6 line 12: It should be Fig. 4E, not 4F

Done. Thanks.

7) The manuscript contains many Western blots, but I am not sure how often the experiments were repeated. That

should be clearly written in the captions. They were not quantified. While, at large, that might not be necessary in every case, in Fig. 6C that should be done as the difference between lanes 3 and 9 is not easily visible and the conclusion (p.8 line 1) is therefore in this case not convincing. Sometimes the blots are overexposed, like in Fig 2J (pS166), Fig 4B (FLAG-PPP1R3G, PP1gamma) or 4C (RIPK1 and PP1gamma), Fig 6C (MLKL). Why was 14-3-3 used in some cases for loading control, not a more commonly used loading control like actin, tubulin or GAPDH?

Quantification has been done for the mentioned blots. And many mentioned blots have been replaced. We used 14-3-3 as a loading control, mainly because it is an abundant protein that runs below 34 kDa. So it will not interfere with other Western blotting results after stripping some of the membranes.

8) Fig. 5B and C shows effects of compound treatment in PPP1R3G-KO cells only, based on the assumption that PP1 activity will not be required to activate cell death if the phosphorylation is already inhibited. The comparison to WT is lacking, it should be the same according to their theory. Particularly because in Fig. 5D, where the experiment is done based on the previous findings, then the authors use WT cells, which is then not comparable to the previous experiments.

Results of HT-29 cells treated with the compounds are shown in the revised Fig.5B and 5C. These inhibitors did enhance cell death in HT-29 cells, but not to the extent as in PPP1R3G-KO cells.

9) Fig. 5F and 3G mass spectrometry data: The full data needs to be made available via supplementary table.

Please see Supplementary Table 6 and 7.

10) The revision should at least include the source data for the Western blots, if not all source data.

Please see the Supplementary Figure 3.

11) The in vitro experiments are done only with catalytic subunit and not with the holoenzyme. I think that this is acceptable because the isolation of the holoenzyme is not simple, the reaction was controlled by lambda-phosphatase, and PP1gamma was used. Direct dephosphorylation was shown to be possible.

Thanks.

12) P. 7 line 6: "S320 phosphorylation of RIPK1 reached high level at 15 minutes after T/S/Z treatment and decreased dramatically at 4 hours, at which time S166 phosphorylation of RIPK1 was highest (Fig. 5D), confirming previously reported inhibitory role of S320 phosphorylation." I disagree, this only shows a sequential phosphorylation/dephosphorylation, it is not an activity assay. It should be rephrased.

This is a valid point. This part has been rewritten.

13) In the last chapter on the knockout mouse, more information on the mouse model and the disease model is necessary to understand this fully. How does the SIRS model work? For example, the sentences from the discussion could be moved here: "Ppp1r3g-deficient mice are viable and have less glycogen deposition in liver. They also have reduced body weight and fat composition upon high-fat diet feeding⁵³." At the moment it sounds like they were the first ones to create the knockout model. Also "SIRS is an excessive defense response of the body to insults, including infection, trauma, acute inflammation, and ischemia. TNF-induced SIRS is a mouse model of sterile sepsis⁴³, widely used in recent times to evaluate the in vivo function of necroptosis pathway components." can be moved into the mouse results chapter from the discussion.

Thanks for the suggestions. This part has been rewritten.

14) Fig. 7: Why do the authors look at ileum and liver? In Fig. 7c ileum should be shown in addition to liver because both tissues were looked at with H&E staining, and the protein levels should also be compared between wt and knockout tissues.

Ileum and liver have been reported to be severely damaged in TNF-induced SIRS (reference 50 and 51). RT-PCR in ileum was shown in the revised Fig. 7C. Western blotting of liver lysates is shown in revised Fig. 7J.

15) Fig. 2C caption: why does PPP1R3G run at 55kDa? Is this known or how do the authors know what they are looking at?

We mentioned in the figure legend of Fig. 2C that PPP1R3G runs on SDS-PAGE gel at around 55kDa, which is similar to what was reported in reference 45. In addition, bacteria produced GST-PPP1R3G ran at about 80 kDa, suggesting that the discrepancy between the predicted molecular weight (37.5 kDa) and the position on the gel (55 KDa) was not due to modifications in mammalian cells.

REVIEWER COMMENTS

Reviewer #1 (Remarks to the Author):

The authors made multiple changes to the manuscript, which significantly improved it. I appreciate the authors emphasis on distinct types of Complex I, which I think makes the model much clearer. I am still not entirely convinced in the sequence of events in the upstream complexes in the presence of SMAC mimetics as it relates to inhibitory phosphorylation of RIPK1, because such phosphorylation requires RIPK1 ubiquitination. It is possible that LUBAC may exclusively function in this capacity in the absence of K63 ubiquitination and/or E3 ligases besides cIAPs operate in Complex I. I don't think c-Cbl is a good example in this case, as it is inducing pro-death ubiquitination downstream from phosphorylation/dephosphorylation steps in Complex I. Therefore, c-Cbl function is quite different from cIAP1/2. My concerns are more reflective of the knowledge gaps in the field, rather than specific critiques of the manuscript. Overall, I think the revised manuscript is generally acceptable for the publication. However, because the model depends on RIPK1 phosphorylation in Complex I in response to T/S/Z but not T/CHX/Z, I would really like to see experiment in Fig. 4C performed under T/CHX/Z conditions to insure that we don't see p-Ser166 RIPK1 in complex I under these conditions in the author's hands.

Reviewer #2 (Remarks to the Author):

Major points:

1) We asked the authors to present how negative control guides behaved in their genome wide CRISPR screen, in order to provide insight on the levels of noise in the screen. The authors responded to this by simply extracting the negative control guides from the overall library and providing them as Supp. Table 3. However, it seems that they have not used the controls in any meaningful analyses. We plotted this ourselves and the negative controls clearly show a large spread, similar to the entire library, indicating a large amount of noise in the screen.

Notably, such a result does not invalidate the entire study, but provides the reader with useful context for the signal strength in the screen. Further, the negative control guides can be randomly subsetted into 'dummy' genes, to determine an empirical false positive rate.

2) We asked the authors to include information on the number of replicates and a demonstration of replicate reproducibility. The authors did not perform the screen in replicates but instead compared Set A to Set B of the GeCKOV2 library. In Supplementary Figure 1, the authors performed principal component analysis on the two sets and found the Spearman correlation p-value significance. This is a very nontraditional approach and does not provide much insight into how the screen performed. Furthermore, it is unclear if this analysis was done on all data or only the “most significant guide for each gene.” Instead, the data should be presented as a scatterplot and a measurement of correlation should be provided. Additionally, the genes that the authors further explore should be annotated in this scatterplot. Below we have provided a sample figure based on the provided data - please provide a similar scatterplot in place of the current Supplementary Figure 1. Please include the performance of the ‘dummy’ genes, each comprised of 3 randomly-chosen negative control guides from each set, in such a plot.

3) We asked the authors to provide more information for the siRNA screen. They provided the sequences in Supplementary Table 5 and the impact of the siRNAs on cell viability in Supplementary Figure 2, but the screening data should also be included in the table.

Reviewer #4 (Remarks to the Author):

The authors have addressed my comments largely satisfactorily.

2 remaining comments:

1. It would help the understanding of the work to explain a bit the potential reasons for the differences between WT and KO cells in figures 5B and C, which have not been discussed yet.
2. I noticed that supp. fig 2 is upside-down.

Reviewer #5 (Remarks to the Author):

The authors have successfully addressed all the mass spectrometry related previously raised concerns and thus I am pleased to recommend this manuscript for publication.

Here are our point-by-point responses to the reviewers' comments which are reiterated in italics.

Reviewer #1:

The authors made multiple changes to the manuscript, which significantly improved it. I appreciate the authors emphasis on distinct types of Complex I, which I think makes the model much clearer. I am still not entirely convinced in the sequence of events in the upstream complexes in the presence of SMAC mimetics as it relates to inhibitory phosphorylation of RIPK1, because such phosphorylation requires RIPK1 ubiquitination. It is possible that LUBAC may exclusively function in this capacity in the absence of K63 ubiquitination and/or E3 ligases besides cIAPs operate in Complex I. I don't think c-Cbl is a good example in this case, as it is inducing pro-death ubiquitination downstream from phosphorylation/dephosphorylation steps in Complex I. Therefore, c-Cbl function is quite different from cIAP1/2. My concerns are more reflective of the knowledge gaps in the field, rather than specific critiques of the manuscript. Overall, I think the revised manuscript is generally acceptable for the publication. However, because the model depends on RIPK1 phosphorylation in Complex I in response to T/S/Z but not T/CHX/Z, I would really like to see experiment in Fig. 4C performed under T/CHX/Z conditions to insure that we don't see p-Ser166 RIPK1 in complex I under these conditions in the author's hands.

The reviewer raised a good point. There is still a lot to learn about the regulation of RIPK1 activation in complex I. Reference 15 (PMID: 29891719) reported that T/CHX/Z treatment did not induce detectable S166 phosphorylation of RIPK1 until 4 hrs post treatment, while it was first induced at 15' with T/5Z-7/Z treatment. As shown in the new Fig 4C, 1 hr post T/S/Z treatment, p-S166-RIPK1 was present in the input and anti-FLAG-TNF IP sample (lane 3 and lane 11), while no p-S166-RIPK1 was detected in T/CHX/Z-treated samples (lane 4 and lane 12), confirming that RIPK1 activation is not required in complex I for T/CHX/Z-induced necroptosis.

Reviewer #2:

1) We asked the authors to present how negative control guides behaved in their genome wide CRISPR screen, in order to provide insight on the levels of noise in the screen. The authors responded to this by simply extracting the negative control guides from the overall library and providing them as Supp. Table 3. However, it seems that they have not used the controls in any meaningful analyses. We plotted this ourselves and the negative controls clearly show a large spread, similar to the entire library, indicating a large amount of noise in the screen.

Notably, such a result does not invalidate the entire study, but provides the reader with useful context for the signal strength in the screen. Further, the negative control guides can be randomly subsetted into 'dummy' genes, to determine an empirical false positive rate.

We agree that negative control shows some spread similar to that of the entire library, indicative of noise in the screen. But we did not intend to make CRISPR screen the purpose of the study and it is just a method to find genes of interest. It led us to concentrate on PPP1R3G. That fulfilled the purpose of the screen.

As per reviewer's suggestion, we used the negative control guides in a new analysis. We first randomly subsetted the negative control guides into dummy genes. We then used MAGECK test command with input options "control-sgrna option" and "norm-method control" (https://sourceforge.net/p/mageck/wiki/QA/#what-does-the-control-sgrna-control_sgrna-option-do-how-to-use-this-option). MAGECK uses the list of negative control guides to generate the null distribution of RRA score during p-value calculation. As per the recommendation from authors of MAGECK, we used 1000 negative control guides for p-value estimation. The gene summary results show an improvement in the p-value calculation and reduction in noise. However, in the end, the top positively selected genes irrespective of the analysis methods remain similar and PPP1R3G is still in the top 10.

without input option: control-sgrna				with input option: control-sgrna			
rank	id	p-value	good sgRNA	rank	id	p-value	good sgRNA
1	MLKL	2.27E-07	5	1	MLKL	2.36E-07	5
2	RIPK1	2.27E-07	4	2	RIPK1	2.36E-07	4
3	RIPK3	2.27E-07	4	3	RIPK3	2.36E-07	4
4	MLL4	6.81E-07	5	4	MLL4	7.08E-06	4
5	TNFRSF1A	6.81E-07	3	5	MAP2K7	1.01E-05	3
6	MAP2K7	1.14E-06	3	6	TNFRSF1A	6.68E-05	2
7	CHUK	5.68E-06	4	7	PPP1R3G	7.34E-05	2
8	SPINT1	5.68E-06	5	8	CHUK	0.00016	3
9	PPP1R3G	7.95E-06	3	9	RAB27A	0.000197	3
10	FBN1	1.25E-05	4	10	C6orf201	0.00023	3

2) We asked the authors to include information on the number of replicates and a demonstration of replicate reproducibility. The authors did not perform the screen in replicates but instead compared Set A to Set B of the GeCKOV2 library. In Supplementary Figure 1, the authors performed principal component analysis on the two sets and found the Spearman correlation p-value significance. This is a very nontraditional approach and does not provide much insight into how the screen performed. Furthermore, it is unclear if this analysis was done on all data or only the “most significant guide for each gene.” Instead, the data should be presented as a scatterplot and a measurement of correlation should be provided. Additionally, the genes that the authors further explore should be annotated in this scatterplot. Below we have provided a sample figure based on the provided data - please provide a similar scatterplot in place of the current Supplementary Figure 1. Please include the performance of the ‘dummy’ genes, each comprised of 3 randomly-chosen negative control guides from each set, in such a plot.

The screen was done several years ago, and we did not perform the screen in replicates. Again, the purpose of the screen was to find genes of interest and it did lead us to PPP1R3G. As per reviewer’s suggestions we have presented the data as a scatter plot. In the plot we have annotated the dummy genes (corresponding to negative control guides) and also highlighted genes of interest. Briefly, for GeCKOV2 library set A and B, fold change between control and T/S/Z sample was calculated for each gene using MAGeCK and then plotted. Spearman correlation coefficient and significance of correlation is shown in the plot.

In addition, we have submitted two files to GEO. (1) FASTQ files; and (2) Median normalized counts of sgRNA generated by MAGeCK (reference 63). Please retrieve the files using the following link, Go to <https://www.ncbi.nlm.nih.gov/geo/query/acc.cgi?acc=GSE176422> Enter token opwvimaibzwp tet into the box.

3) We asked the authors to provide more information for the siRNA screen. They provided the sequences in Supplementary Table 5 and the impact of the siRNAs on cell viability in Supplementary Figure 2, but the screening data should also be included in the table.

We have included the cell viability results in the new supplementary table 5.

Reviewer #4:

The authors have addressed my comments largely satisfactorily.

2 remaining comments:

1. It would help the understanding of the work to explain a bit the potential reasons for the differences between WT and KO cells in figures 5B and C, which have not been discussed yet.

We thank the reviewer for the comments. This part has been rewritten as requested.

2. I noticed that supp. fig 2 is upside-down.

It was set at landscape orientation to accommodate the content of the graph. The fonts will be too small if it is set at portrait orientation.

Reviewer #5:

The authors have successfully addressed all the mass spectrometry related previously raised concerns and thus I am pleased to recommend this manuscript for publication.

Thanks.

REVIEWERS' COMMENTS

Reviewer #1 (Remarks to the Author):

The authors satisfactory addressed my concern. I recommend manuscript for acceptance.

Reviewer #2 (Remarks to the Author):

I have no further comments for these authors.